# V1T: large-scale mouse V1 response prediction using a Vision Transformer

**Bryan M. Li**[1]                                                                 *bryan.li@ed.ac.uk*

**Isabel M. Cornacchia**[1]                                                        *isabel.cornacchia@ed.ac.uk*

**Nathalie L. Rochefort**[2,3]                                                     *n.rochefort@ed.ac.uk*

**Arno Onken**[1]                                                                  *aonken@ed.ac.uk*

[1]*School of Informatics, University of Edinburgh*
[2]*Centre for Discovery Brain Sciences, University of Edinburgh*
[3]*Simons Initiative for the Developing Brain, University of Edinburgh*

**Reviewed on OpenReview:** *https://openreview.net/forum?id=qHZs2p4ZD4*

## Abstract

Accurate predictive models of the visual cortex neural response to natural visual stimuli remain a challenge in computational neuroscience. In this work, we introduce V1T, a novel Vision Transformer based architecture that learns a shared visual and behavioral representation across animals. We evaluate our model on two large datasets recorded from mouse primary visual cortex and outperform previous convolution-based models by more than 12.7% in prediction performance. Moreover, we show that the self-attention weights learned by the Transformer correlate with the population receptive fields. Our model thus sets a new benchmark for neural response prediction and can be used jointly with behavioral and neural recordings to reveal meaningful characteristic features of the visual cortex. Code available at github.com/bryanlimy/V1T.

## 1 Introduction

Understanding how the visual system processes information is a fundamental challenge in neuroscience. Predictive models of neural responses to naturally occurring stimuli have shown to be a successful approach toward this goal, serving the dual purpose of generating new hypotheses about biological vision (Bashivan et al., 2019; Walker et al., 2019; Ponce et al., 2019) and bridging the gap between biological and computer vision (Li et al., 2019; Sinz et al., 2019; Safarani et al., 2021). This approach relies on the idea that high performing predictive models, which explain a large part of the stimulus-driven variability, have to account for the nonlinear response properties of the neural activity, thus allowing for the identification of the underlying computations of the visual system (Carandini et al., 2005).

An extensive amount of work on the primary visual cortex (V1) has been dedicated to building quantitative models that accurately describe neural responses to visual stimuli, starting from simple linear-nonlinear models (Heeger, 1992; Jones and Palmer, 1987), energy models (Adelson and Bergen, 1985) and multi-layer models (Lehky et al., 1992; Lau et al., 2002; Prenger et al., 2004). These models, based on neurophysiological data, provide a powerful framework to test hypotheses about neural functions and investigate the principles of visual processing. With the increased popularity of deep neural networks (DNNs) in computational neuroscience in recent years (Kietzmann et al., 2018; Richards et al., 2019; Li et al., 2020; 2021), DNNs have set new standards of prediction performance (Antolík et al., 2016; Klindt et al., 2017; Ecker et al., 2018; Zhang et al., 2019), allowing for a more extensive exploration of the underlying computations in sensory processing (Walker et al., 2019; Bashivan et al., 2019; Burg et al., 2021).

DNN-based models are characterized by two main approaches. On the one hand, task-driven models rely on pre-trained networks optimized on standard vision tasks, such as object recognition, in combination with a readout mechanism to predict neural responses (Yamins et al., 2014; Cadieu et al., 2014; Cadena et al., 2019). With the goal of explaining the evolutionary and developmental constraints of the visual system, task-driven models have proven to be successful for predicting visual responses in primates (Yamins and DiCarlo, 2016; Cadena et al., 2019) and mice (Nayebi et al., 2022) by obtaining a shared generalized representation of the visual input across animals. On the other hand, data-driven models aim to build a predictive model on large-scale datasets without any assumption on the functional properties of the network. These models share a common representation by being trained end-to-end directly on data from thousands of neurons, and they have been shown to be successful as predictive models for the mouse visual cortex (Lurz et al., 2021; Franke et al., 2022). This approach allows us to identify core components that can be insightful when studying nontrivial computational properties of cortical neurons, especially in combination with experimental verification (Walker et al., 2019).

Data-driven models for prediction of visual responses across multiple animals typically employ the core-readout framework (Klindt et al., 2017; Cadena et al., 2019; Lurz et al., 2021; Burg et al., 2021; Franke et al., 2022). Namely, a core module which learns a shared latent representation of the visual stimuli across the animals, followed by animal-specific linear readout modules to predict neural responses given the latent features. This architecture enforces the nonlinear computations to be performed by the shared core, which can in principle capture general characteristic features of the visual cortex (Lurz et al., 2021). The readout models then learn the animal-specific mapping from the shared representation of the input to the individual neural responses. With the advent of large-scale neural recordings, datasets that consist of thousands or even hundreds of thousands of neurons are becoming readily available (Stosiek et al., 2003; Steinmetz et al., 2021). This has led to an increase in the number of parameters needed in the readout network to account for the large number of neurons, hence significant effort in neural predictive modeling has been dedicated to develop more efficient readout networks. On the other hand, due to their effectiveness and computation efficiency (Goodfellow et al., 2016), convolutional neural networks (CNNs) are usually chosen as the shared representation model.

Recently, Vision Transformer (ViT, Dosovitskiy et al. 2021) has achieved excellent results in a broad range of computer vision tasks (Han et al., 2022) and Transformer-based (Vaswani et al., 2017) models have become increasingly popular in computational neuroscience (Tuli et al., 2021; Schneider et al., 2022; Whittington et al., 2022). For instance, Ye and Pandarinath (2021) proposed a Neural Data Transformer to model spike trains, which was extended by Le and Shlizerman (2022) using a Spatial Transformer to achieve state-of-the-art performance in 4 neural datasets. Berrios and Deza (2022) introduced a data augmentation and adversarial training procedure to train a dual-stream Transformer which showed strong performance in predicting monkey V4 responses. In modeling the mouse visual cortex, Conwell et al. (2021) experimented with a wide range of out-of-the-box DNNs, including CNNs and ViTs, to compare their representational similarity when pre-trained versus randomly initialized. Here, we explore the benefits of the ViT convolution-free approach and self-attention mechanism as the core representation learner in a data-driven neural predictive model. Note that, in this text, the term "attention" strictly refers to the self-attention layer in Transformers (Vaswani et al., 2017), which is distinct from the perceptual process of "attention" in the neuroscience literature.

Since neural variability shows a significant correlation with the internal brain state (Pakan et al., 2016; 2018; Stringer et al., 2019), information about behavior can greatly improve visual system models in the prediction of neural responses (Bashiri et al., 2021; Franke et al., 2022). To exploit this relationship, we also investigate a principled mechanism in the model architecture to integrate behavioral states with visual information.

Altogether, we propose V1T, a novel ViT-based architecture that can capture visual and behavioral representations of the mouse visual cortex. This core architecture, in combination with an efficient per-animal readout (Lurz et al., 2021), outperforms the previous state-of-the-art model by 12.7% and 19.1% on two large-scale mouse V1 datasets (Willeke et al., 2022; Franke et al., 2022), which consist of neural recordings of thousands of neurons across over a dozen behaving rodents in response to thousands of natural images. Moreover, we show that the attention weights learned by the core module correlate with behavioral variables, such as pupil direction. This link between the model and the visual cortex activity is useful for pinpointing how behavioral variables affect neural activity.

## 2    Neural data

We considered two large-scale neural datasets for this work, DATASET S[1] by Willeke et al. (2022) and DATASET F by Franke et al. (2022). These two datasets consist of V1 recordings from behaving rodents in response to thousands of natural images, providing an excellent platform to evaluate our proposed method and compare it against previous visual predictive models.

We first briefly describe the animal experiment in DATASET S. A head-fixed mouse was placed on a cylindrical treadmill with a 25 inch monitor placed 15 cm away from the animal's left eye and more than 7,000 neurons from layer L2/3 in V1 were recorded via two-photon calcium imaging. Note that the position of the monitor was selected such that the stimuli were shown to the center of the recorded population receptive field. Gray-scale images $x_{\text{image}} \in \mathbb{R}^{c=1 \times h \times w}$ from ImageNet (Deng et al., 2009) were presented to the animal for 500 ms with a blank screen period of 300 to 500 ms between each presentation. Neural activities were accumulated between 50 and 500 ms after each stimulus onset. In other words, for a given neuron $i$ in trial (stimulus presentation) $t$, the neural response is represented by a single value $r_{i,t}$. In addition, the anatomical coordinates of each neuron as well as four behavioral variables $x_{\text{behaviors}}$ were recorded alongside with the calcium responses. These variables include pupil dilation, the derivative of the pupil dilation, pupil center (2d-coordinates) and running speed of the animal. Each recording session consists of up to 6,000 image presentations (i.e. trials), where 5,000 unique images are combined with 10 repetitions of 100 additional unique images, randomly intermixed. The 1,000 trials with repeated images are used as the test set and the rest are divided into train and validation sets with a split ratio of 90% and 10% respectively. In total, data from 5 rodents[2] (MOUSE A to E) were recorded in this dataset.

DATASET F follows largely the same experimental setup with the following distinction: colored images (UV-colored and green-colored, i.e. $x_{\text{image}} \in \mathbb{R}^{c=2 \times h \times w}$) from ImageNet were presented on a screen placed 12 cm away from the animal; 4,500 unique colored and 750 monochromatic images were used as the training set and an additional 100 unique colored and 50 monochromatic images were repeated 10 times throughout the recording; in total, 10 rodents (MOUSE F to O) were used in the experiment with 1,000 V1 neurons recorded from each animal. Table A.1 summarizes the experimental information from both datasets.

## 3    Previous work

A substantial body of work has recently focused on predictive models of cortical activity that learn a shared representation across neurons (Klindt et al., 2017; Cadena et al., 2019; Lurz et al., 2021; Burg et al., 2021; Franke et al., 2022), which stems from the idea in systems neuroscience that cortical computations share common features across animals (Olshausen and Field, 1996). In DNN models, these generalizing features are learned in a nonlinear core module, then a subsequent neuron-specific readout module linearly combines the relevant features in this representation to predict the neural responses. Recently, Lurz et al. (2021) and Franke et al. (2022) introduced a shared CNN core and animal-specific Gaussian readout combination that achieved excellent performance in mouse V1 neural response prediction, and this is the current state-of-the-art model on large-scale benchmarks including DATASET S and DATASET F. Here, we provide a brief description for each of the modules in their proposed architecture, which our work is built upon.

**CNN core**. Typically, the core module learns the shared visual representation via a series of convolutional blocks (Cadena et al., 2019; Lurz et al., 2021; Franke et al., 2022). In Lurz et al. (2021), given an input image $x_{\text{image}} \in \mathbb{R}^{c \times h \times w}$, the CNN core with filter size $k$ outputs a latent representation vector $z \in \mathbb{R}^{d \times h' \times w'}$ where $h' = h - k + 1$, $w' = w - k + 1$ and $d$ is the hidden dimension. The CNN core, after an exhaustive Bayesian hyperparameter search to optimize for the validation performance, has an output dimension of $z \in \mathbb{R}^{d \times h'=28 \times w'=56}$. Previous works have shown correlation between behaviors and neural variability, and that the behavioral variables can significantly improve neural predictivity (Niell and Stryker, 2010; Reimer et al., 2014; Stringer et al., 2019; Bashiri et al., 2021). To that end, Franke et al. (2022) proposed to integrate the behavioral variables $x_{\text{behaviors}} \in \mathbb{R}^v$ with the visual stimulus by duplicating each variable to a $h \times w$ matrix and concatenating them with $x_{\text{image}}$ in the channel dimension, resulting in an input vector of $\mathbb{R}^{(c+v) \times h \times w}$.

---

[1]The Sensorium Challenge held at NeurIPS 2022 Competition Track Program

[2]2 additional mice were used in the Sensorium challenge (Willeke et al., 2022) and their test sets are not publicly available.

**Readout**. To compute the neural response of neuron $i$ from mouse $m$ with $n_m$ neurons, the readout module $\mathtt{R}_m : \mathbb{R}^{d \times h' \times w'} \to \mathbb{R}^{n_m}$ by Lurz et al. (2021) computes a linear regression of the core representation $z$ with weights $w_i \in \mathbb{R}^{w' \times h' \times c}$, followed by an ELU activation with an offset of 1 (i.e. $o = \mathrm{ELU}(\mathtt{R}_m(z)) + 1$), which keeps the response positive. The regression is performed by a Gaussian readout, which learns the parameters of a 2d Gaussian distribution whose mean $\mu_i$ represents the center of the receptive field of the neuron in the image space and whose variance quantifies the uncertainty of the receptive field position, which decreases over training. The response is thus obtained as a linear combination of the feature vector of the core at a single spatial position, which allows the model to greatly reduce the number of parameters per neuron in the readout. Notably, to learn the position $\mu_i$, the model also exploits the retinotopic organization of V1 by coupling the recorded cortical 2d coordinates of each neuron with the estimated center of the receptive field from the readout. Moreover, a shifter module is introduced to adjust (or shift) the $\mu_i$ receptive field center of neuron $i$ to account for the trial-to-trial variability due to eye movement (Franke et al., 2022). The shifter network $\mathbb{R}^2 \to \mathbb{R}^2$ consists of 3 dense layers with hidden size of 5 and tanh activation; it takes as input the 2d pupil center coordinates and learns the vertical and horizontal adjustments needed to shift $\mu_i$.

# 4 Methods

The aim of this work is to design a neural predictive model $F(x_{\mathrm{image}}, x_{\mathrm{behaviors}})$ that can effectively incorporate both visual stimuli and behavioral variables to predict responses $o$ that are faithful to real recordings $r$ from mouse V1. With that goal, we first detail the core architectures proposed in this work, followed by the training procedure and evaluation metrics. Code used in this work is attached as supplementary material and will be made publicly available upon publication.

## 4.1 V1T core

Vision Transformers (Dosovitskiy et al., 2021), or ViTs, have achieved competitive performance in many computer vision tasks, including object detection and semantic segmentation, to name a few (Chen et al., 2020; Carion et al., 2020; Strudel et al., 2021). Here, we propose a data-driven ViT core capable of learning a shared representation of the visual stimuli that is relevant for the prediction of neural responses in the visual cortex. Moreover, we introduce an alternative approach in V1T to encode behavioral variables in a more principled way when compared to previous methods and further improve the neural predictive performance of the overall model.

The original ViT classifier is comprised of 3 main components: (1) a tokenizer first encodes the 3d image (including channel dimension) into 2d patch embeddings, (2) the embeddings are then passed through a series of Transformer (Vaswani et al., 2017) encoder blocks, each consisting of a Multi-Head Attention (`MHA`) and a Multi-Layer Perceptron (`MLP`) module which requires 2d inputs, and finally (3) a classification layer outputs the class prediction. The following sections detail the modifications made to convert the vanilla ViT to a shared visual representation learner for the downstream readout modules. We additionally experiment with a number of recently proposed efficient ViTs that have been emphasized for learning from small to medium size datasets.

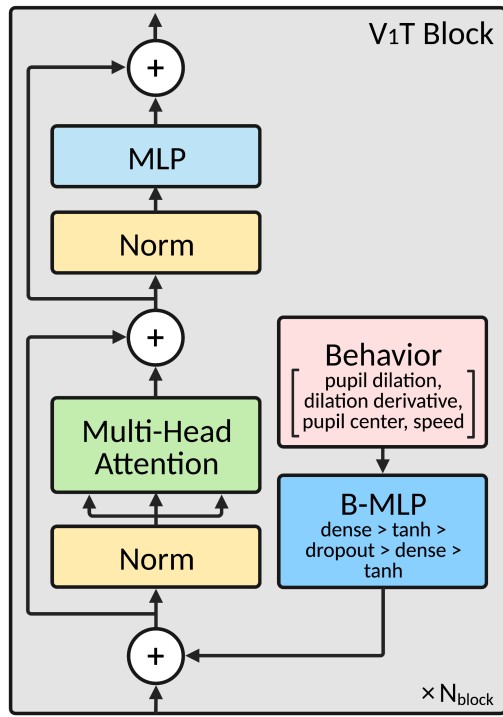

Figure 1: Illustration of the V1T block architecture.

**Tokenizer**. The tokenizer, or patch encoder, extracts non-overlapping squared patches of size $p \times p$ from the 2d image and projects each patch to embeddings $z_0$ of size $d$, i.e. $\mathbb{R}^{c \times h \times w} \rightarrow \mathbb{R}^{l \times (cp^2)} \rightarrow \mathbb{R}^{l \times d}$, where $l = hw/p^2$ is the number of patches. Dosovitskiy et al. (2021) proposed two tokenization methods in the original ViT, where patches can be extracted either (1) via a $p \times p$ sliding window over the height and width dimensions of the image, followed by a linear layer with $d$ hidden units, or (2) via a 2d convolutional layer with kernel size $p$ and $d$ filters.

Transformer-based models benefit from (or even necessitate) pre-training on large datasets, in the magnitude of millions or even billions of samples, in order to obtain optimal performance (Han et al., 2022). In contrast, typical neural recordings in animal experiments are considerably smaller. To stay consistent with previous work, we instead focus on developing a core architecture that can be effectively trained on limited amount of data from scratch. To that end, we considered two recently introduced efficient ViT methods that are highly competitive in scarce data settings. Lee et al. (2021) proposed Shifted Patch Tokenization (SPT) to combat the low inductive bias in ViTs and enable better learning from limited data. Conceptually, SPT allows additional (adjacent) pixel values to be included in each patch, thus improving the locality, or receptive field, of the model. Input image $x_{\text{image}} \in \mathbb{R}^{1 \times h \times w}$ is shifted spatially by $p/2$ in one of the four diagonal directions (top-left, top-right, bottom-left, bottom-right) with zero padding and the four shifted images (i.e. each shifted in one diagonal direction) are then concatenated with the original image, resulting in a vector $\mathbb{R}^{5 \times h \times w}$, which can be processed by the two patch extraction approaches mentioned above. With a similar goal in mind, the Compact Convolutional Transformer (CCT, Hassani et al. 2021) was proposed as a convolutional tokenizer to learn the patch embeddings that can take advantage of the translation equivariance and locality inherent in CNNs. The proposed mini-CNN is fairly simple: it consists of a 2d convolution layer with a $p \times p$ kernel and filter size $d$, followed by ReLU activation and a max pool layer. In this work, we experimented with and compared all four tokenization methods: sliding window, a single 2d convolutional layer, SPT and CCT.

As ViTs are agnostic to the spatial structure of the data, a positional embedding is added to each patch to encode the relative position of the patches with respect to each other (Dosovitskiy et al., 2021; Han et al., 2022) and this positional embedding can either be learned or sinusoidal. Finally, a learnable BERT (Devlin et al., 2019) `[cls]` token is typically added to the patch embeddings (i.e. $z_0 \in \mathbb{R}^{(l+1) \times d}$) to represent the class of the image.

**Transformer encoder**. The encoder consists of a series of ViT blocks, where each block comprises two sub-modules: Multi-Head Attention (`MHA`) and Multi-Layer Perceptron (`MLP`). In each `MHA` module, we applied the standard self-attention formulation (Vaswani et al., 2017): $\text{Attention}(Q, K, V) = \text{softmax}(QK^T/\sqrt{d})V$, where query $Q$, key $K$ and value $V$ are linear projections of the input $z_b$ at block $b$. Conceptually, the self-attention layer assigns a pairwise attention value among all the patches (or tokens). In addition to the standard formulation, we also experimented with the Locality Self Attention (LSA, Lee et al. 2021), where a diagonal mask is applied to $QK^T$ to prevent strong connections in self-tokens (i.e. diagonal values in $QK^T$), thus improving the locality inductive bias. Each sub-module is preceded by Layer Normalization (`LayerNorm`, Ba et al. 2016), and followed by a residual connection to the next module.

**Reshape representation**. To make the dimensions compatible with the Gaussian readout module (see Section 3 for an overview), we reshape the 2d core output $z \in \mathbb{R}^{l \times d}$ to $\mathbb{R}^{d \times h' \times w'}$, where $l = h' \times w'$ and $h' \leq w'$. Note that if the number of patches $l$ is not sufficiently large, it is possible for the same position in $z$ to be mapped to multiple neurons, which could lead to adverse effects. For instance, in the extreme case of $l = 1$, all neurons would be mapped to a single $p \times p$ region in the visual stimulus (i.e. they would have the same visual receptive field), which is not biologically plausible given the size of the recorded cortical area (Garrett et al., 2014). We therefore set the stride size of the patch encoder as a hyperparameter and allow for overlapping patches, thus letting the hyperparameter optimization algorithm select the optimal number of patches. Given $x_{\text{image}} \in \mathbb{R}^{c \times h = 36 \times w = 64}$, the V1T core has an output dimension of $\mathbb{R}^{d \times h' = 29 \times w' = 57}$.

### 4.1.1 Incorporating behaviors

Previous studies have shown that visual responses can be influenced by behavioral variables and brain states; for example, changes in arousal, which can be monitored by tracking pupil dilation, lead to stronger (or weaker) neural responses (Reimer et al., 2016; Larsen and Waters, 2018). As a consequence, the visual

representation learned by the core module should also be adjusted according to the brain state. Here, instead of inputting a vector that is a concatenation of the visual stimulus $x_{\text{image}} \in \mathbb{R}^{c \times h \times w}$ and behavioral information $x_{\text{behaviors}} \in \mathbb{R}^v$ in the channel dimension (i.e. $\mathbb{R}^{(c+v) \times h \times w}$, see Section 3), we propose an alternative method to integrate behavioral variables with the visual stimulus using a novel ViT-based architecture – V1T, illustrated in Figure 1.

We introduced a behavior MLP module ($\texttt{B-MLP} : \mathbb{R}^v \to \mathbb{R}^d$) at the beginning of the encoder block which learns to adjust the visual latent vector $z$ based on the observed behavioral states $x_{\text{behaviors}}$. Each $\texttt{B-MLP}$ module comprises two fully-connected layers with $d$ hidden units and a dropout layer in between; tanh activation is used so that the adjustments to $z$ can be both positive and negative. Importantly, as layers in DNNs learn different features of the input, usually increasingly abstract and complex with deeper layers (Zeiler and Fergus, 2014; Raghu et al., 2021), we hypothesize that the influence of the internal brain state should therefore change from layer to layer. To that end, we learned a separate $\texttt{B-MLP}_b$ at each block $b$ in the V1T core, thus allowing level-wise adjustments to the visual latent variable. Formally, $\texttt{B-MLP}_b$ projects $x_{\text{behaviors}}$ to the same dimension of the embeddings $z_{b-1}$, followed by an element-wise summation between latent behavioral and visual representations, and then the rest of the operations in the encoder block:

$$z_b \leftarrow z_{b-1} + \texttt{B-MLP}_b(x_{\text{behaviors}}) \tag{1}$$

$$z_b \leftarrow \texttt{MHA}_b(\texttt{LayerNorm}(z_b)) + z_b \tag{2}$$

$$z_b \leftarrow \texttt{MLP}_b(\texttt{LayerNorm}(z_b)) + z_b \tag{3}$$

where $z_0$ denotes the original patch embeddings. To compare the prediction performance difference due to our proposed behavior module, we also trained an equivalent Vision Transformer (denoted as ViT) with the same architecture as V1T except that it integrates behavioral information in the same manner as the CNN model (i.e. ViT inputs $\mathbb{R}^{(c+v) \times h \times w}$).

## 4.2 Training and evaluation

In order to isolate the change in prediction performance that is solely due to the proposed core architectures, we employed the same readout architectures by Lurz et al. (2021), as well as a similar data preprocessing and model training procedure. We used the same train, validation and test split provided by the two datasets (see Section 2). Natural images, recorded responses, and behavioral variables (i.e. pupil dilation, dilation derivative, pupil center, running speed) were standardized using the mean and standard deviation measured from the training set and the images were then resized to $36 \times 64$ pixels from $144 \times 256$ pixels. The shared core and per-animal readout modules were trained jointly using the AdamW optimizer (Loshchilov and Hutter, 2019) to minimize the Poisson loss

$$\mathcal{L}_m^{\text{Poisson}}(r, o) = \sum_{t=1}^{n_t} \sum_{i=1}^{n_m} \left( o_{i,t} - r_{i,t} \log(o_{i,t}) \right) \tag{4}$$

between the recorded responses $r$ and predicted responses $o$, where $n_t$ is the number of trials in one batch and $n_m$ the number of neurons for mouse $m$. A small value $\varepsilon = 1e-8$ was added to both $r$ and $o$ prior to the loss calculation to improve numeric stability. Gradients from each mouse were accumulated before a single gradient update to all modules. We tried to separate the gradient update for each animal, i.e. one gradient update per core-readout combination, but this led to a significant drop in performance. We suspect this is because the core module failed to learn a generalized representation among all animals when each update step only accounted for gradient signals from one animal. We used a learning rate scheduler in conjunction with early stopping: if the validation loss did not improve over 10 consecutive epochs, we reduced the learning rate by a factor of 0.3; if the model still had not improved after 2 learning rate reductions, we then terminated the training process. Dropout (Srivastava et al., 2014), stochastic depths (Huang et al., 2016), and L1 weight regularization were added to prevent overfitting. The weights in dense layers were initialized by sampling from a truncated normal distribution ($\mu = 0.0, \sigma = 0.02$), where the bias values were set to 0.0; whereas the weight and bias in $\texttt{LayerNorm}$ were set to 1.0 and 0.0. Each model was trained on a single Nvidia RTX 2080Ti GPU and all models converged within 200 epochs. Finally, we employed Hyperband Bayesian optimization (Li et al., 2017) to find the hyperparameters that achieved the best performance in the validation set. This

included finding the optimal tokenization method and self-attention mechanism. The initial search space and final hyperparameter settings are detailed in Table A.2. We independently performed a hyperparameter search on the CNN model, though we failed to find a configuration that achieves better performance than the settings provided by Lurz et al. (2021) and Franke et al. (2022). While learning rate warm-up and pre-training on large datasets are considered the standard approach to train Transformers (Xiong et al., 2020; Han et al., 2022), in order to stay consistent with previous work and to isolate the performance gain solely due to the architectural change, all models presented in this work are trained from scratch and follow the same procedure stated above.

The prediction performance of our models was measured by the single trial correlation metric, used by Willeke et al. (2022) and Franke et al. (2022), which can also account for the trial-to-trial variability in the test set where the same visual stimuli were shown multiple times. We computed the correlation between recorded $r$ and predicted $o$ responses:

$$\text{trial corr.}(r, o) = \frac{\sum_{i,j}(r_{i,j} - \bar{r})(o_{i,j} - \bar{o})}{\sqrt{\sum_{i,j}(r_{i,j} - \bar{r})^2 \sum_{i,j}(o_{i,j} - \bar{o})^2}} \tag{5}$$

where $\bar{r}$ and $\bar{o}$ are the average recorded and predicted responses across all trials in the test set.

## 5 Results

Here, we first discuss the final core architecture chosen after the Bayesian hyperparameter optimization, followed by a comparison of our proposed core against baseline models on the two large-scale mouse V1 datasets. Moreover, we analyze the trained core module and present the insights that can be gained from it. We present the cross-animal and cross-dataset generalization in Appendix A.4.

Table 1: The single trial correlation (CORR.) between predicted and recorded responses in DATASET S and DATASET F test set. $\Delta$CNN and $\Delta$ViT show the relative differences against the CNN (Lurz et al., 2021) and ViT models with behavior variables; we additionally fitted a CNN and ViT core with stimulus-response pairs (BEHAV: ✗) to evaluate the prediction performance without behavioral information. SD shows the standard deviation across animals and detailed per-animal results are available in Appendix A.3.

|  | BEHAV | DATASET S (WILLEKE ET AL.) | | | DATASET F (FRANKE ET AL.) | | |
|---|---|---|---|---|---|---|---|
|  |  | CORR. (SD) | $\Delta$CNN | $\Delta$ViT | CORR. (SD) | $\Delta$CNN | $\Delta$ViT |
| LN | ✓ | 0.275 (0.019) | -27.2% | -33.7% | 0.223 (0.040) | -28.0% | -35.4% |
| CNN | ✗ | 0.300 (0.021) | -20.6% | -27.6% |  |  |  |
| CNN | ✓ | 0.378 (0.029) | 0.0% | -8.7% | 0.309 (0.070) | 0.0% | -10.3% |
| ViT | ✗ | 0.319 (0.024) | -15.6% | -22.9% |  |  |  |
| ViT | ✓ | 0.414 (0.032) | +9.5% | 0.0% | 0.344 (0.041) | +11.4% | 0.0% |
| V1T | ✓ | **0.426** (0.027) | +12.7% | +3.0% | **0.368** (0.032) | +19.1% | +6.9% |
| ENSEMBLE OF 5 MODELS | | | | | | | |
| CNN | ✓ | 0.404 (0.025) | +6.9% | -2.3% | 0.340 (0.050) | +10.0% | -1.3% |
| ViT | ✓ | 0.424 (0.026) | +12.2% | +2.4% | 0.365 (0.037) | +18.1% | +6.0% |
| V1T | ✓ | **0.439** (0.027) | +16.1% | +6.1% | **0.378** (0.033) | +22.3% | +3.8% |

**V1T benefits from smaller and overlapping patches**. We first looked at how hyperparameters of ViT and V1T affect model performance. We observed the predictive performance to be quite sensitive towards number of patches, patch size and patch stride. The most performant models used a patch size of 8 and a stride size of 1, thus extracting the maximum number of patches. We note that this allows the readout to learn a mapping from the shared core representation of the stimulus to the cortical position of each neuron that spans across the whole image, and not just a part of the image. Since the visual receptive fields of neurons are distributed across a large area of the monitor given the size of the recorded cortical area, this leads to more accurate response predictions from the model. Furthermore, we found that the two efficient

tokenizers, SPT and CCT, whose aim is to reduce the number of patches, both failed to improve the model performance, reiterating that a finer tiling of the image is crucial for accurate predictions of cortical activity. Moreover, we found that the LSA attention mechanism, which encourages the model to learn from inter-tokens by masking out the diagonal self-token, led to worse performance, suggesting information from adjacent patches in this task is not as influential as it is in image classification. Appendix A.1 details the importance of each hyperparameter and the test performance trade-off among the various tokenizers and attention mechanisms. Lastly, we found that V1T with layer-wise `B-MLP` modules yields the best results, indicating that the modulation introduced by behavioral information varies as the core learns different visual representations at deep layers. Further analysis and discussion on the `B-MLP` module are presented in Appendix A.2.

**V1T outperforms CNN**. Next, we compared the tuned ViT and V1T cores against a baseline linear-nonlinear (LN) model and the previous state-of-the-art CNN model (Lurz et al., 2021) on the two large scale mouse V1 datasets (see Section 2). We also trained a CNN and ViT core on response-stimuli pairs only on DATASET S, to evaluate the importance of behavioral information in response predictions. Table 1 summarizes the test performance on the two datasets, results of per-animal performance and an alternative metric are available in Appendix A.3. By simply replacing the CNN core module with the tuned ViT architecture, we observed a considerable improvement in response predictions across all animals, with an average increase of 9.5% and 11.3% in single trial correlation over the CNN model in DATASET S and DATASET F respectively. Thus far, the core module encoded the brain state of the animals by concatenating behavioral variables as additional channels in the natural image. With that said, our proposed V1T core, which encodes the brain state via the `B-MLP` nonlinear transformations, further improved the average prediction performance by 2.9% and 7.0% in the two datasets, or 12.7% and 19.1% over the CNN model.

As demonstrated in the Sensorium Challenge (Sensorium Workshop, 2022) and Franke et al. (2022), ensemble learning is a common approach to improve neural predictive models. Following the procedure in Franke et al. (2022), we trained 10 models with different random seeds and selected the 5 best models based on their validation performance. The average of the selected models constituted the output of the ensemble model. The CNN ensemble model achieved an average improvement of 6.9% in DATASET S as compared to its non-ensemble variant. Nevertheless, the individual V1T model still outperformed the CNN ensemble by 5.4%. A V1T ensemble trained with the same procedure achieved an average single trial correlation of 0.439, which corresponds to an 8.7% improvement over the CNN ensemble model. Altogether, our proposed core architecture set a new benchmark in both gray-scale and colored visual response prediction.

**Sample efficiency**. Most neural datasets are constrained by their limited size, due to technical and/or ethical limitations, while typical DNNs require a large amount of data to train on, especially Transformer-based models (Han et al., 2022). Here, we evaluate the sample efficiency of the CNN, ViT and V1T models by fitting them with 500 (11%), 1500 (33%), 2500 (55%), 3500 (77%) and 4500 (100%) samples per animal in DATASET S (Willeke et al., 2022). Figure 2 shows the single trial correlation in the test set for the three models trained on different sample sizes, each with 30 different random seeds. Overall, we found that V1T outperforms the CNN model even at 1500 training samples per animal. Moreover, the predictive performance of the CNN model plateaus at around 3500 training samples, while V1T keeps improving, suggesting that the ViT-based model can continue to improve with more data.

**Spatial tuning difference**. As expected, models trained without behavioral information led to worse results (see BEHAVIOR: ✗ in Table 1). Nevertheless, we observed an average 6.3% improvement in stimuli-response prediction with the tuned ViT core over the CNN model in DATASET S. To further our understanding of why the ViT might be performing better in visual response prediction, we evaluated the discrepancies in spatial tuning of the two models by comparing their artificial receptive fields (aRFs). Appendix A.5 details the procedure. Briefly, we presented the models with thousands of white noise images and then summed the images weighted by the response prediction to estimate the aRF of each artificial unit. Figure 3a shows the aRF of the same artificial unit from the CNN and ViT model. Visually, the aRFs of the ViT model appear to be narrower and qualitatively different from the aRFs of the CNN. In order to quantify the aRF sizes, we fitted a 2d Gaussian to each aRF and observed a significant difference in the standard deviation distributions, shown in Figure 3b. Overall, the aRFs of the ViT model have a much narrower spread, with a mean standard deviation of $3.0 \pm 0.5$ and $2.6 \pm 0.4$ in the horizontal and vertical directions over all artificial units, considerably lower than the $5.1 \pm 1.5$ and $3.1 \pm 0.9$ of the CNN. These results show that the artificial

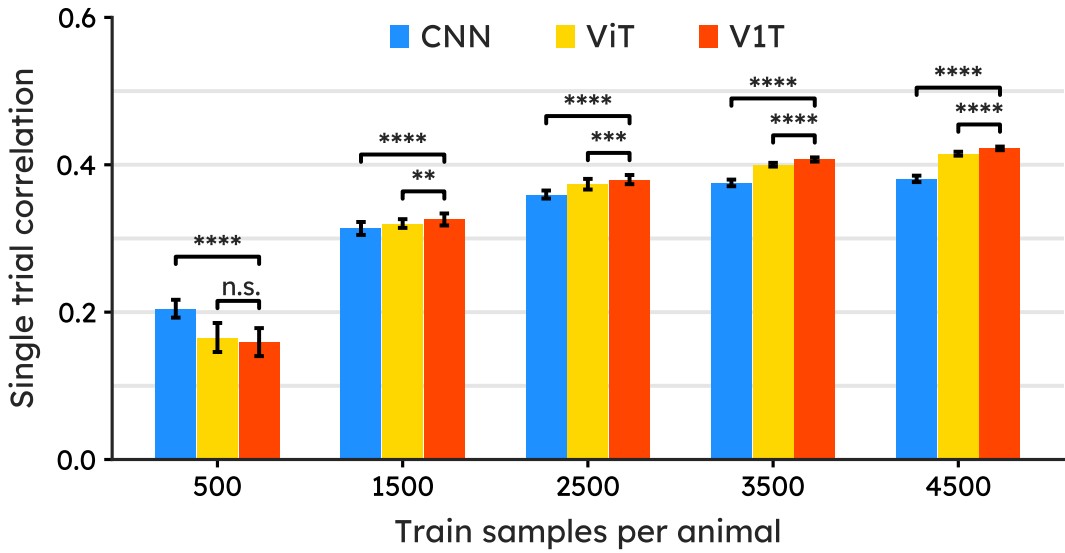

Figure 2: Prediction performance when trained with 500, 1500, 2500, 3500 and (all) 4500 samples per animal in DATASET S. The models were each trained with 30 different random seeds. The error bar shows the standard deviations of the repeated experiments, and the statistical difference (two-sided t-test) in CNN vs V1T and ViT vs V1T in each sample group is shown above each pair of bars (****: $p \leq 0.0001$).

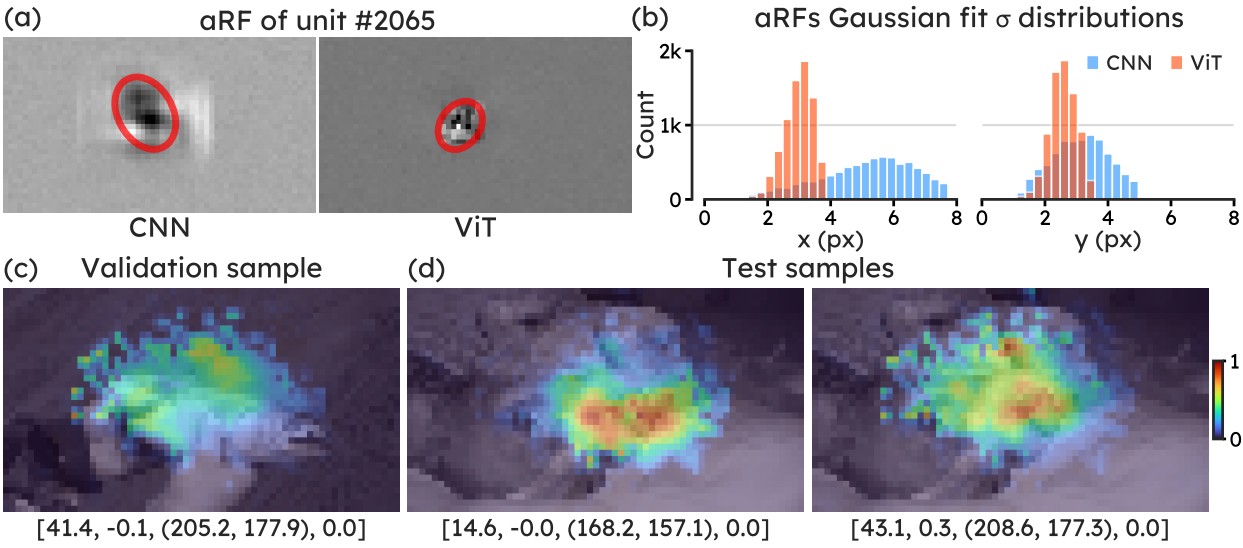

Figure 3: (a) Estimated artificial receptive field (aRF) and 2d Gaussian fit (red circle shows 1 standard deviation ellipse) of the same artificial unit from the CNN and ViT models trained without behaviors. Visually, the ViT learns narrower aRFs, more examples in Appendix A.5. To quantify the size of the aRFs, we compared the fitted Gaussian over all units from MOUSE A; (b) the distributions of the standard deviations shows that the ViT learns notably narrower aRFs. V1T attention visualization on MOUSE A (c) validation and (d) test samples. Each attention map was normalized to [0, 1], and the behavioral variables of the corresponding trial are shown below the image in the format of [pupil dilation, dilation derivative, pupil center $(x, y)$, speed]. More examples in Appendix A.6.

units in the CNN and ViT learn notably different aRFs. Given that we did not constrain the aRF size, our results suggest that the narrower fields allow ViT to learn location-dependent features that are beneficial for visual response prediction.

**Self-attention visualization**. In addition to the performance gain in the proposed core architecture, the self-attention mechanism inherent in Transformers can be used to visualize areas in the input image that the model learns to focus on. In our case, it allows us to detect the regions in the visual stimulus that drive the neural responses. To that end, we extracted the per-stimulus attention map learned by the V1T core module via Attention Rollout (Abnar and Zuidema, 2020; Dosovitskiy et al., 2021). Briefly, we aggregated the attention weights (i.e. $\text{Softmax}(QK^T/\sqrt{d})$) across all heads in MHA, and then multiplied the weights over all layers (blocks), recursively. Figure 3 shows the normalized average attention weights superimposed to the input images from Mouse A, with more examples available in Appendix A.6. Given that the position of the computer monitor was chosen in order to center the population receptive field, V1 responses from the recorded region should be mostly influenced by the center of the image (Willeke et al., 2022). Here, we can see a clear trend where the core module is focusing on the central regions of the images to predict the neural response, which aligns with our expectation from the experiment conditions. Interestingly, when the core module inputs the same image but with varying behaviors (i.e. Figure 3d), we noticed variations in the attention patterns. This suggests that the V1T core is able to take behavioral variables into consideration and adjust its attention solely based on the brain state.

These attention maps can inform us of the area of the image (ir)relevant for triggering visual neuronal responses which, in turn, allow us to build more sophisticated predictive models. For instance, the core module consistently assigned higher weights to patches in the center of the image, suggesting information at the edges of the image are less (or not at all) relevant for the recorded group of neurons. As a practical example, we eliminated irrelevant information in the stimuli by center cropping the image to $\alpha 144 \times \alpha 256$ pixels where $0 < \alpha \leq 1$, prior to downsampling the input to $36 \times 64$ pixels. We found that a crop factor of $\alpha = 0.8$ (i.e. removing 36% of the total number of pixels) further improved the single trial correlation to 0.430, or 13.8% better than the CNN. Note that we also obtained similar improvement with the CNN model.

**Self-attention correlates with pupil center**. To further explore the relationship between the attention weights learned by the core module and the behavioral information, we measured the absolute correlation between the center of mass of the attention maps and the pupil centers in the vertical and horizontal axes. The correlation coefficient of each animal in Dataset S is summarized in Table 2. Overall, we found a moderate correlation between the attention maps and the pupil center of the animal, with an average correlation (standard deviation) of 0.525 (0.079) and 0.409 (0.105) in the horizontal and vertical directions across animals. This relationship demonstrates that the attention maps can reveal the impact of behavioral variables on the neural responses. Therefore, this framework can be particularly useful for studies investigating the coding of visual information across visual cortical areas (V1 and higher visual areas), as the model could determine what part(s) of the visual stimulus is processed along the "hierarchy" of visual cortical areas. Since higher visual areas are known to have larger receptive fields (Wang and Burkhalter, 2007; Glickfeld et al., 2014), we would expect a larger part of the image to be relevant for the core module. Further investigation of the attention map could also be used to determine which part of a visual scene was relevant when performing more specific tasks, such as object recognition, decision-making, or spatial navigation.

Table 2: Correlations between the center of mass of the attention maps and pupil centers in the (x-axis) horizontal and (y-axis) vertical direction in Dataset S test set, all with a p-value $\ll 0.0001$.

| Mouse | X-axis | Y-axis |
|-------|--------|--------|
| A | 0.682 (****) | 0.568 (****) |
| B | 0.489 (****) | 0.493 (****) |
| C | 0.505 (****) | 0.370 (****) |
| D | 0.484 (****) | 0.310 (****) |
| E | 0.464 (****) | 0.302 (****) |

## 6 Discussion

In this work, we presented a novel core architecture V1T to model the visual and behavioral representations of mouse V1 activities in response to natural visual stimuli. The model outperformed the previous state-of-the-art CNN (Lurz et al., 2021) model on two large-scale mouse V1 datasets by a considerable margin (12.7% and 19.1%). In contrast to the winning submissions at the Sensorium Challenge (Sensorium Workshop, 2022), which focused on data augmentation and building large ensembles based on the CNN model, we instead introduced a new architecture as the shared core module. Our best model achieved a single trial correlation of 0.428 and 0.444 (correlation to average: 0.634 and 0.650) in the two held-out test sets, which would place us 2nd place in the leaderboard, and the best method across all models not taking the neuronal response trends over time into account. In addition, we also showed that V1T can be competitive in the low data regime, and that its performance continues to improve with more data to a larger extend than the CNN model. To the best of our knowledge, our approach is also the first ViT-based model to outperform CNNs in mouse V1 response prediction.

With a strong neural predictive performance, this model also provides a framework to investigate *in silico* the computations in the visual system, and in particular, the modulation of neural responses by behavioral variables. In this study, we included speed of the animal in the virtual corridor, pupil dilation, dilation derivative and pupil center as behavioral variables. For each of these variables, there is prior evidence showing that they do affect responses in V1. For instance, Pakan et al. (2018) showed that 12% of the recorded V1 neurons decreased their activity with lower running speed, suggesting a clear benefit of considering the speed of the animal for predicting V1 responses. Pupil dilation has been shown to be related to arousal of the animal, with complex modality dependent effects of arousal on the mouse sensory cortex (Shimaoka et al., 2018). The pupil center represents the fixation point of the animal and is a proxy for what the animal is paying attention to. As a proof of principle of how a Vision Transformer can be used to gain insights into the importance of behavioral variables for V1 responses, we showed that the center of the self-attention maps learned by our model correlates with the pupil center of the animals, highlighting how features of this architecture do reflect properties of cortical neurons' receptive fields, in this case, the retinotopy. Moreover, our model is able to exploit certain anatomical information, for example the location of neurons within the primary visual cortex, from which we can roughly infer the location of their receptive field since the retinotopic map of mouse primary visual cortex is well characterized (Zhuang et al., 2017). However, while the CNN architecture was inspired by receptive fields of the visual cortex (Fukushima, 1980), the Vision Transformer architecture was not and has no direct biological counterpart. Therefore, it is challenging to map the abstract components of a Vision Transformer onto the anatomy or biophysics of the brain.

Nevertheless, the V1T model has a number of limitations. Firstly, only one-dimensional behavioral information can be incorporated since the model integrates scalars into the latent embedding via the `B-MLP` module. Additional architecture engineering is needed if the behavioral variables have varying (and higher) dimensions, for instance, 3D poses (Mathis et al., 2018). Secondly, in the case of very limited data (e.g. 500 samples, see Figure 2), CNN-based models are likely to outperform ViTs, which typically require considerable amount of data to be performant (Han et al., 2022).

In future work, we plan to further investigate the relationship between behavioral variables and neural responses. The attention visualization technique, for instance, enables ablation studies on the effect of each behavioral variable, such as pupil dilation or running speed, on the neural activity. Moreover, we plan to extend the method to recordings of the visual cortex in response to natural videos, to track how this relationship may evolve over time, as well as experiments in naturalistic settings, to know which part of a visual scene is relevant for certain behaviors.

**Acknowledgments**

We sincerely thank Willeke et al. and Franke et al. for making their high-quality large-scale mouse recordings publicly available which make this work possible. We would also like to thank Antonio Vergari, Matthias Hennig and Robyn Greene for their insightful comments and suggestions on improving the manuscript. BML was supported by the United Kingdom Research and Innovation (grant EP/S02431X/1), UKRI Centre for Doctoral Training in Biomedical AI at the University of Edinburgh, School of Informatics. This project has

received funding from the European Research Council (ERC) under the European Union's Horizon 2020 research and innovation programme (grant agreement No. 866386; to N.R.). For the purpose of open access, the author has applied a creative commons attribution (CC BY) licence to any author accepted manuscript version arising.

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

# A  Appendix

Table A.1: Experimental information of Mouse A to E from Dataset S (Willeke et al., 2022) and Mouse F to O from Dataset F (Franke et al., 2022). Each mouse has a unique recording ID (column 2) although we assigned a separate mouse ID (column 1) to use throughout this paper for simplicity.

| Mouse | rec. ID | num. neurons | total trials | num. test |
|-------|---------|--------------|--------------|-----------|
| A | 21067-10-18 | 8372 | 5994 | 998 |
| B | 22846-10-16 | 7344 | 5997 | 999 |
| C | 23343-5-17 | 7334 | 5951 | 989 |
| D | 23656-14-22 | 8107 | 5966 | 993 |
| E | 23964-4-22 | 8098 | 5983 | 994 |
| F | 25311-10-26 | 867 | 7358 | 1475 |
| G | 25340-3-19 | 922 | 7478 | 1497 |
| H | 25704-2-12 | 773 | 7500 | 1500 |
| I | 25830-10-4 | 1024 | 7360 | 1473 |
| J | 26085-6-3 | 910 | 7464 | 1495 |
| K | 26142-2-11 | 1121 | 7500 | 1500 |
| L | 26426-18-13 | 1125 | 7500 | 1500 |
| M | 26470-4-5 | 1160 | 7473 | 1495 |
| N | 26644-6-2 | 824 | 7500 | 1500 |
| O | 26872-21-6 | 1109 | 7466 | 1495 |

## A.1 Hyperparameters

Table A.2: ViT and V₁T cores - Gaussian readout hyperparameter search space and their final settings after a Hyperband Bayesian optimization (Li et al., 2017).

| HYPERPARAMETER | SEARCH SPACE | FINAL VALUE |
|---|---|---|
| CORE | | |
| NUM. BLOCKS | UNIFORM, MIN: 1, MAX: 8 | 4 |
| NUM. HEADS | UNIFORM, MIN: 1, MAX: 12 | 4 |
| PATCH SIZE | UNIFORM, MIN: 2, MAX: 16 | 8 |
| PATCH STRIDE | UNIFORM, MIN: 1, MAX: PATCH SIZE | 1 |
| PATCH METHOD | SLIDING WINDOW, 2D CONV, SPT, CCT | SLIDING WINDOW |
| PATCH DROPOUT | UNIFORM, MIN: 0, MAX: 0.5 | 0.0229 |
| EMBEDDING SIZE | UNIFORM, MIN: 8, MAX: 1024, INTERVAL: 1 | 155 |
| MHA METHOD | ORIGINAL, LSA | ORIGINAL |
| MHA DROPOUT | UNIFORM, MIN: 0, MAX: 0.5 | 0.2544 |
| MLP SIZE | UNIFORM, MIN: 8, MAX: 1024, INTERVAL: 1 | 488 |
| MLP DROPOUT | UNIFORM, MIN: 0, MAX: 0.5 | 0.2544 |
| STOCHASTIC DEPTH DROPOUT | UNIFORM, MIN: 0, MAX: 0.5 | 0.0 |
| L1 WEIGHT REGULARIZATION | UNIFORM, MIN: 0, MAX: 1 | 0.5379 |
| INITIAL LEARNING RATE | UNIFORM, MIN: 0.005, MAX: 0.0001 | 0.0016 |
| READOUT | | |
| POSITION NETWORK NUM. LAYERS | UNIFORM, MIN: 1, MAX: 4, INTERVAL: 1 | 1 |
| POSITION NETWORK NUM. UNITS | UNIFORM, MIN: 2, MAX: 128, INTERVAL: 2 | 30 |
| BIAS INITIALIZATION | 0, MEAN STANDARDIZED RESPONSE | 0 |
| L1 WEIGHT REGULARIZATION | UNIFORM, MIN: 0, MAX: 1 | 0.0076 |

Table A.3: ViT and V₁T hyperparameter importance in Hyberhand Bayesian Optimization (Li et al., 2017) via Weights & Biases (Biewald, 2020). IMPORTANCE shows the degree to which the hyperparameter is useful to predict the evaluation metric (e.g. single trial correlation in the validation set) and COR-RELATION shows the linear correlation between the hyperparameter and the evaluation metric. Details on the calculation and interpretation of the hyperparameter importance and correlation are available at docs.wandb.ai/guides/app/features/panels/parameter-importance.

| HYPERPARAMETER | IMPORTANCE | CORRELATION |
|---|---|---|
| EMBEDDING SIZE | 0.393 | -0.626 |
| PATCH STRIDE | 0.164 | -0.358 |
| PATCH SIZE | 0.111 | -0.297 |
| INITIAL LEARNING RATE | 0.046 | 0.279 |
| L1 WEIGHT REGULARIZATION | 0.030 | -0.242 |
| NUM. BLOCKS | 0.030 | 0.093 |
| NUM. HEADS | 0.028 | -0.070 |
| BATCH SIZE | 0.026 | -0.093 |
| MHA DROPOUT | 0.025 | -0.034 |
| PATCH METHOD | 0.024 | -0.174 |
| MLP DROPOUT | 0.022 | 0.133 |
| MLP SIZE | 0.019 | -0.186 |
| STOCHASTIC DEPTH DROPOUT | 0.019 | -0.225 |
| PATCH DROPOUT | 0.017 | -0.105 |
| MHA METHOD | 0.014 | 0.001 |

Table A.4: Best prediction performance in single trial correlation (standard deviation across animals) on Dataset S with respect to choice of attention formulation and patch/tokenization method. ORIGINAL denotes the original self-attention formulation by Vaswani et al. 2017 and LSA denotes the Locality Self Attention mechanism proposed by Lee et al. 2021. SPT denotes Shifted Patch Tokenization (Lee et al., 2021) and CCT denotes the tokenization method introduced in Compact Convolution Transformer (Hassani et al., 2021). Section 4.1 details the model architectural differences and Section 5 discusses their prediction results.

| MHA \ PATCH METHOD | SLIDING WINDOW | 2D CONV | SPT | CCT |
|---|---|---|---|---|
| ORIGINAL | **0.426** (0.027) | 0.411 (0.022) | 0.406 (0.024) | 0.392 (0.026) |
| LSA | 0.413 (0.023) | 0.415 (0.024) | 0.405 (0.024) | 0.385 (0.025) |

## A.2 `B-MLP` activation

We investigated different variations of the `B-MLP` module. The motivation of the proposed behavior module is to enable the core to learn a shared representation of the visual and behavioral variables across the animals. Moreover, the level-wise connections allow the self-attention module in each V1T block to encode different behavioral features with the latent visual representation. We experimented with a per-animal `B-MLP` module (while the rest of the core was still shared across animals) which did not perform any better than the shared counterpart, suggesting that the behavior module can indeed learn a shared internal brain state presentation. We also tested having the module in the first block only, as well as using the same module across all blocks (i.e. all $B\text{-}MLP_b$ shared the same weights). Both cases, however, led to worse results with a $2-4\%$ reduction in predictive performance on average. To further examine the proposed formulation, we analyzed the activation patterns of the shared behavior module at each level in V1T, shown in Figure A.1. We observed a noticeable distinction in `B-MLP` outputs in earlier versus deeper layers, with a higher spread in deeper layers, which corroborates our hypothesis that the influence of the behavioral variables differs at each level of the visual representation process.

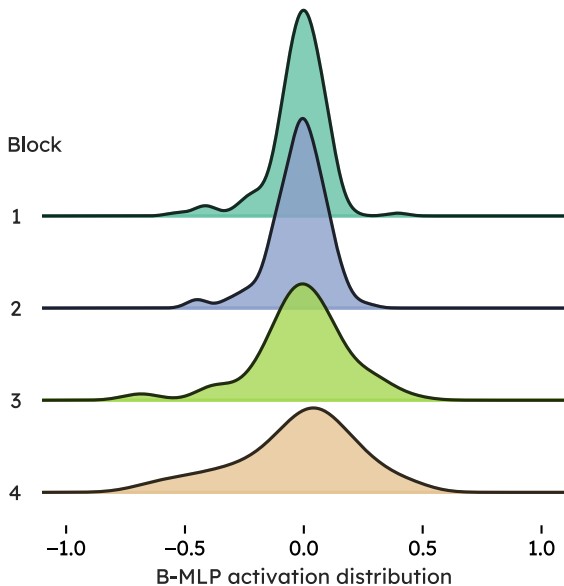

Figure A.1: tanh activation distributions of `B-MLP` at each level (block) in the V1T core. The spread of activation distributions indicates varying influence of behavioral variables at the block in the core module.

### A.3 Prediction results

Table A.5: Single trial correlation between predicted and recorded responses in DATASET S test set. All models were trained with behaviors. To demonstrate that the extracted attention maps can inform us about the (ir)relevant regions in the visual stimulus, we trained an additional V1T core with images center cropped to $\alpha h \times \alpha w$ pixels (See Section 5).

| | MOUSE | | | | | |
| | A | B | C | D | E | AVG (SD) |
|---|---|---|---|---|---|---|
| LN | 0.262 | 0.306 | 0.281 | 0.263 | 0.262 | 0.275 (0.019) |
| CNN | 0.350 | 0.424 | 0.385 | 0.371 | 0.360 | 0.378 (0.029) |
| ViT | 0.375 | 0.455 | 0.415 | 0.433 | 0.392 | 0.414 (0.032) |
| V1T | 0.401 | 0.464 | 0.430 | 0.436 | 0.401 | 0.426 (0.027) |
| V1T (CENTER CROP $\alpha = 0.8$) | **0.403** | **0.468** | **0.433** | **0.442** | **0.403** | **0.430** (0.028) |
| ENSEMBLE OF 5 MODELS | | | | | | |
| CNN | 0.379 | 0.443 | 0.409 | 0.406 | 0.385 | 0.404 (0.025) |
| ViT | 0.398 | 0.460 | 0.421 | 0.440 | 0.401 | 0.424 (0.026) |
| V1T | **0.414** | **0.475** | **0.443** | **0.452** | **0.413** | **0.439** (0.027) |

Table A.6: Single trial correlation between predicted and recorded responses in DATASET F test set. All models were trained with behaviors.

| | MOUSE | | | | | | | | | | |
| | F | G | H | I | J | K | L | M | N | O | AVG (SD) |
|---|---|---|---|---|---|---|---|---|---|---|---|
| LN | 0.194 | 0.254 | 0.214 | 0.279 | 0.255 | 0.233 | 0.148 | 0.231 | 0.174 | 0.243 | 0.223 (0.040) |
| CNN | 0.253 | 0.371 | 0.184 | 0.377 | 0.329 | 0.319 | 0.207 | 0.331 | 0.341 | 0.376 | 0.309 (0.070) |
| ViT | 0.310 | 0.375 | 0.352 | 0.379 | 0.385 | 0.262 | 0.294 | 0.360 | 0.358 | 0.368 | 0.344 (0.041) |
| V1T | **0.326** | **0.386** | **0.387** | **0.394** | **0.398** | **0.373** | **0.298** | **0.377** | **0.363** | **0.379** | **0.368** (0.032) |
| ENSEMBLE OF 5 MODELS | | | | | | | | | | | |
| CNN | 0.268 | 0.383 | 0.341 | 0.393 | 0.347 | 0.336 | 0.242 | 0.345 | 0.355 | 0.388 | 0.340 (0.050) |
| ViT | 0.321 | 0.384 | 0.363 | 0.404 | 0.406 | 0.374 | 0.302 | 0.385 | 0.323 | 0.387 | 0.365 (0.037) |
| V1T | **0.336** | **0.397** | **0.391** | **0.406** | **0.408** | **0.383** | **0.306** | **0.388** | **0.373** | **0.392** | **0.378** (0.033) |

### A.3.1 Correlation to Average

Correlation to Average (AVG. CORR.) is another commonly used metric to evaluate neural predictive models (Willeke et al., 2022). It is the correlation between $r_{i,j}$ recorded and $o_{i,j}$ predicted responses over repeated $j$ trials of stimulus $i$ :

$$\text{avg. corr.}(r, o) = \frac{\sum_i (\bar{r}_i - \bar{r})(o_i - \bar{o})}{\sqrt{\sum_i (\bar{r}_i - \bar{r})^2 \sum_i (o_i - \bar{o})^2}} \tag{6}$$

where $\bar{r}_i = \frac{1}{J} \sum_{j=1}^{J} r_{i,j}$ is the average response across $J$ repeats, and $\bar{r}$ and $\bar{o}$ are the average recorded and predicted responses across all trials.

Table A.7: The Correlation to Average (AVG. CORR.) between predicted and recorded responses across all animals (SD shows the standard deviation) in DATASET S and DATASET F test sets. Table 1 shows the results in single trial correlation.

| | BEHAV | DATASET S (WILLEKE ET AL.) | | | DATASET F (FRANKE ET AL.) | | |
| | | AVG. CORR. (SD) | ΔCNN | ΔViT | AVG. CORR. (SD) | ΔCNN | ΔViT |
|---|---|---|---|---|---|---|---|
| LN | ✓ | 0.387 (0.023) | -33.1% | -37.7% | 0.312 (0.076) | -39.7% | -42.5% |
| CNN | ✗ | 0.551 (0.024) | -4.7% | -4.6% | | | |
| CNN | ✓ | 0.578 (0.027) | 0.0% | -6.9% | 0.516 (0.142) | 0.0% | -4.7% |
| ViT | ✗ | 0.568 (0.026) | -1.7% | -8.5% | | | |
| ViT | ✓ | 0.621 (0.030) | +7.4% | 0.0% | 0.542 (0.054) | 4.9% | 0.0% |
| V₁T | ✓ | **0.629** (0.029) | +8.9% | 1.4% | **0.551** (0.022) | +6.6% | +1.6% |
| ENSEMBLE OF 5 MODELS | | | | | | | |
| CNN | ✓ | 0.610 (0.027) | +5.5% | -1.7% | **0.567** (0.050) | +9.9% | +4.8% |
| ViT | ✓ | 0.634 (0.027) | +9.7% | +2.1% | 0.566 (0.035) | +9.5% | +4.4% |
| V₁T | ✓ | **0.644** (0.026) | +11.3% | +3.7% | 0.562 (0.023) | +8.9% | +3.8% |

### A.4 Cross-animal and cross-dataset generalization

DNN-based neural predictive models are often neuron/animal specific and do not generalize well to unseen neurons/animals. Here, we evaluate generalization performance of CNN and V1T.

We first tested the cross-animal performance of the CNN and V1T models by performing cross-validation over animals in DATASET S (Willeke et al., 2022). Specifically, we compare the model fitted on one animal (direct setting) against a model that was pre-trained on $N-1$ animals and whose readout was fine-tuned (with core frozen) on the left-out animal (transfer setting). We repeated this process for all 5 animals, and their results are summarized in Table A.8. On average, the V1T model outperformed the CNN model by 3.3% and 6.7% in the direct and transfer settings, respectively. Moreover, the V1T model experienced a larger level of performance gain in the transfer learning setting, with an average prediction improvement of 5.6% over direct training, whereas the CNN had a 2.2% gain. These results suggest that the V1T core can generalize well to unseen animals, and also benefit from transfer learning to a greater extent.

Next, we evaluated the cross-dataset generalization performance. To that end, we fitted the models on the gray-scaled version (average channel dimension) of DATASET F (Franke et al., 2022). We then froze the core module and trained the readouts on DATASET S and compared the loss in performance in this transfer setting. The results are presented in Table A.9 for the two core architectures. We observed a larger performance drop with the frozen V1T model compared to the model trained directly, with an average deficit of $-19.0\%$, versus the $-12.9\%$ drop in the frozen CNN model. Similar to the cross-animal generalization, the CNN model exhibits a higher level of variation in prediction performance over the 5 animals. While the relative performance drop was greater for the V1T core than for the CNN core, V1T achieved better transfer results with an average single trial correlation of 0.345, or about 4.9% better than the frozen CNN (0.329).

Table A.8: **CNN vs V1T cross-animal generalization in Dataset S**. We compare the test performance between (DIRECT) fitting one model per animal and (TRANSFER) pre-training a model on $N-1$ animals and fine-tuning the readout for the $N^{\text{th}}$ animal. We repeat the same leave-one-out process for all animals. $\Delta$DIRECT shows the relatively prediction performance of the TRANSFER models over the DIRECT models.

| | | | MOUSE | | | | |
| | A | B | C | D | E | AVG (SD) | $\Delta$DIRECT |
|---|---|---|---|---|---|---|---|
| **CNN** | | | | | | | |
| DIRECT | 0.332 | 0.422 | 0.389 | 0.400 | 0.335 | 0.376 (0.040) | |
| TRANSFER | 0.357 | 0.420 | 0.386 | 0.398 | 0.359 | 0.384 (0.027) | 2.2% |
| **V1T** | | | | | | | |
| DIRECT | 0.368 | 0.417 | 0.394 | 0.414 | 0.347 | 0.388 (0.030) | |
| TRANSFER | 0.384 | 0.450 | 0.414 | 0.415 | 0.385 | 0.410 (0.027) | 5.6% |

Table A.9: **CNN vs V1T cross-dataset generalization**. We first pre-trained the core module on a gray-scale version of DATASET F, then (TRANSFER) froze the core and fine-tuned the readouts on DATASET S. $\Delta$ORIGINAL shows the test performance drop in the cross-dataset transfer learning setting as compare (ORIGINAL) a model directly trained on DATASET S.

| | | | MOUSE | | | | |
| | A | B | C | D | E | AVG (SD) | $\Delta$ORIGINAL |
|---|---|---|---|---|---|---|---|
| **CNN** | | | | | | | |
| ORIGINAL | 0.350 | 0.424 | 0.385 | 0.371 | 0.360 | 0.378 (0.029) | |
| TRANSFER | 0.314 | 0.353 | 0.337 | 0.316 | 0.327 | 0.329 (0.016) | -12.9% |
| **V1T** | | | | | | | |
| ORIGINAL | 0.401 | 0.464 | 0.430 | 0.436 | 0.401 | 0.426 (0.027) | |
| TRANSFER | 0.327 | 0.382 | 0.347 | 0.343 | 0.328 | 0.345 (0.022) | -19.0% |

### A.5 Artificial receptive fields

Here, we outline the procedure to estimate the artificial receptive fields (aRFs) of the CNN and ViT models (not V1T, since there is no behavior involved) and the process to compare their spatial positions and sizes. We first present each trained model with $N = 500,000$ images of white noise drawn from a uniform distribution. The aRF of unit $i$ is then computed as the summation of all noise images, weighted by the respective output:

$$\text{aRF}_i = \sum_n^N \text{F}(x_n)_i * x_n, \quad x_n \sim \mathcal{U}^{1 \times 36 \times 64} \tag{7}$$

where model $\text{F}$ can be either the CNN or ViT, and $\text{F}(x_n)_i$ denotes the response of unit $i$ given white noise image $x_n$. Figure A.2 shows the estimated aRFs of 3 randomly selected artificial units (out of 8372 in the readout for MOUSE A) from the two models.

To quantify the location and size of the aRFs, we fitted a 2d Gaussian to each aRF and compared the mean and covariance of the fitted parameters. We repeated the same process for all 8372 artificial units. Concretely, we first subtracted the mean from each aRF to center the values on the baseline, then took their absolute values and fitted a 2d Gaussian using SciPy's `curve_fit()` function. Note that not all aRFs have good fit. We thus dropped the bottom 5% of the fitted results. Figure A.2c shows the KDE plot of the fitted Gaussian means from the aRFs of the CNN and ViT. The vast majority of the aRFs are centered with respect to the image, aligning with our expectations from the attention rollout maps (see Section 5). Figure 3b compares the standard deviations in horizontal and vertical direction of the fitted Gaussian.

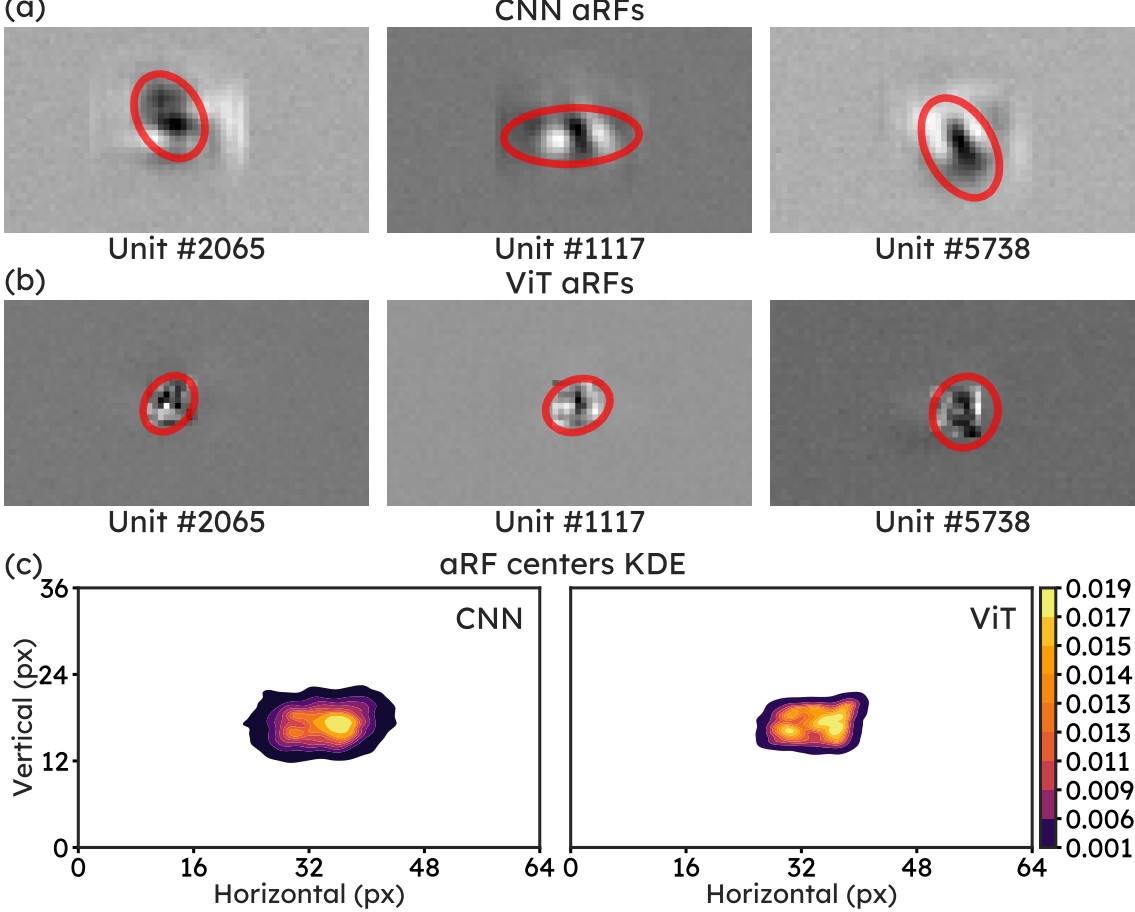

Figure A.2: Estimated artificial receptive fields (aRFs) of (a) CNN and (b) ViT over the same set of randomly selected artificial units from MOUSE A. The red circles (1 standard deviation ellipse) show the 2d Gaussian fit. (c) KDE of the Gaussian centers of the two models.

## A.6 Attention rollout maps

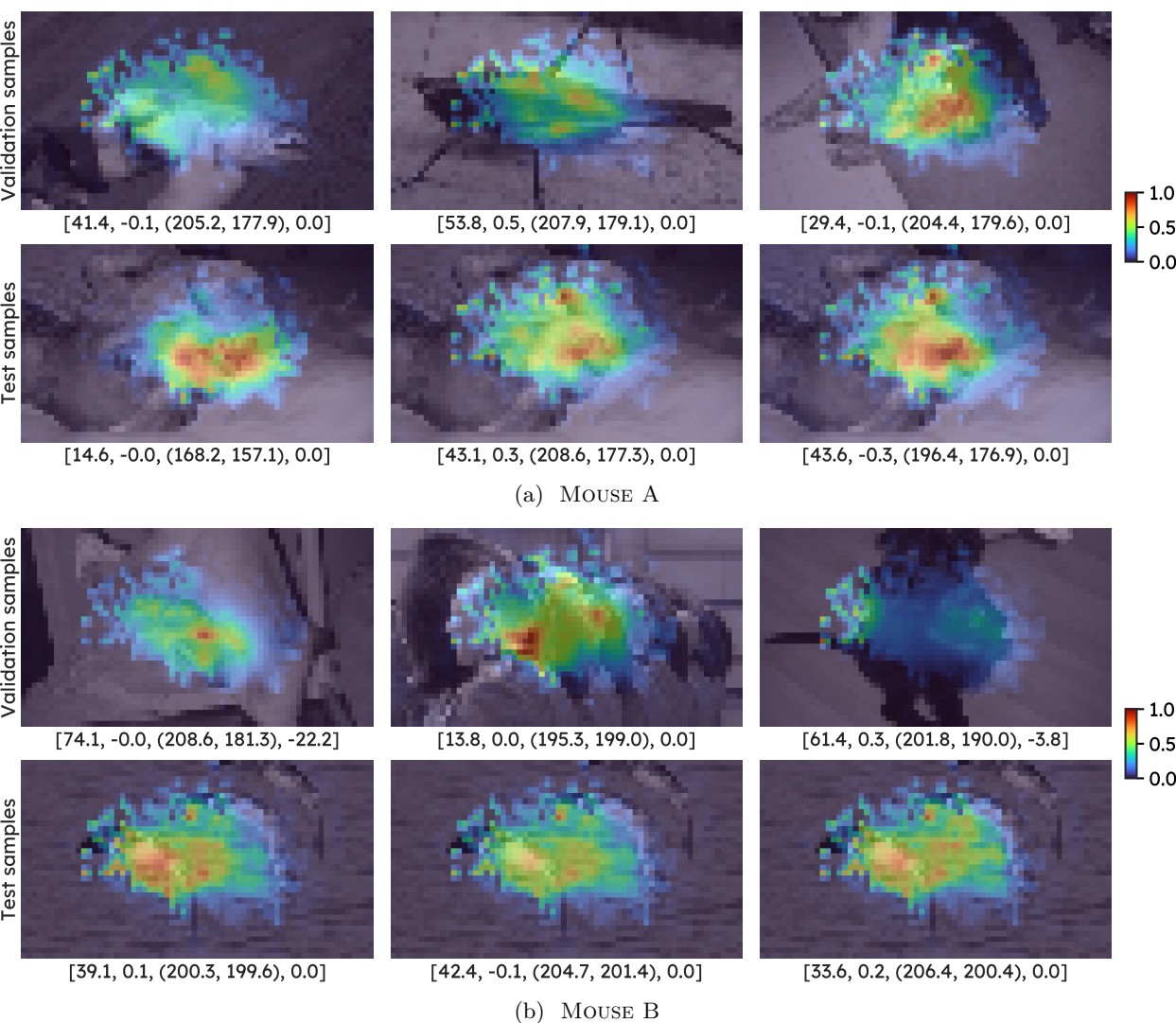

Figure A.3: V1T attention visualization on validation and test samples of MOUSE A to E from DATASET S. As the computer monitor was positioned such that the visual stimuli were presented to the center of the receptive field of the recorded neurons (see DATASET S discussion in Section 2), we expected regions in the center of the image to correlate the most with the neural responses, indicating that the core module learned to assign higher attention weights toward those regions. Note that the core module is shared among all mice. For this reason, we also expected similar patterns across animals. We observed small variations in the attention maps in the test set, where the image is the same and behavioral variables vary, suggesting the core module learned to adjust its attention based on the internal brain state. To quantify this result, we further showed that there are moderate correlations between the center of mass of the attention maps and the pupil center, see discussion in Section 5. Each attention map was normalized to $[0, 1]$, and the behavioral variables of the corresponding trial are shown below the image in the format of [pupil dilation, dilation derivative, pupil center $(x, y)$, speed]. The Figure continues to the next page.

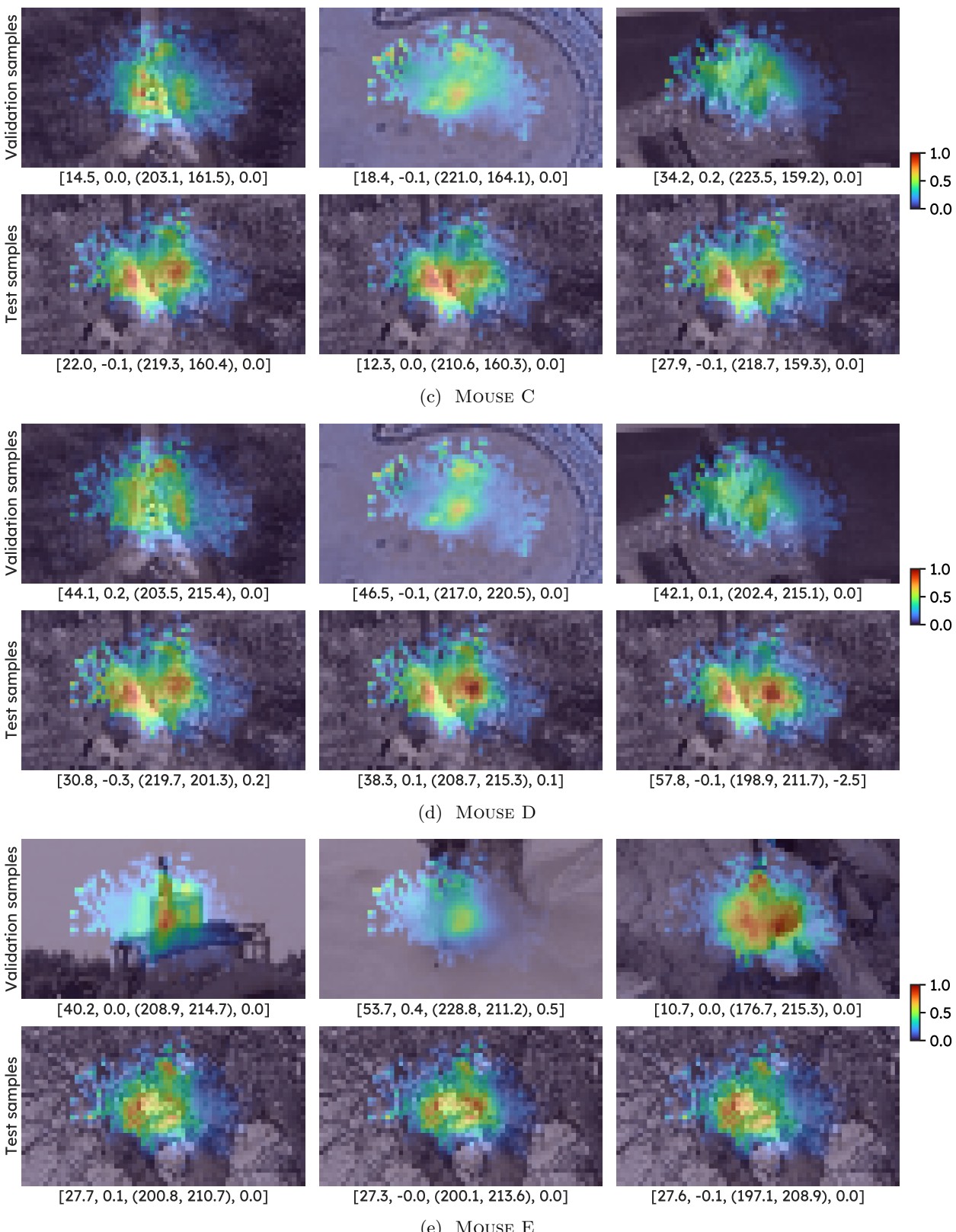

(c) Mouse C

(d) Mouse D

(e) Mouse E

### A.7 Behaviors and predictive performance

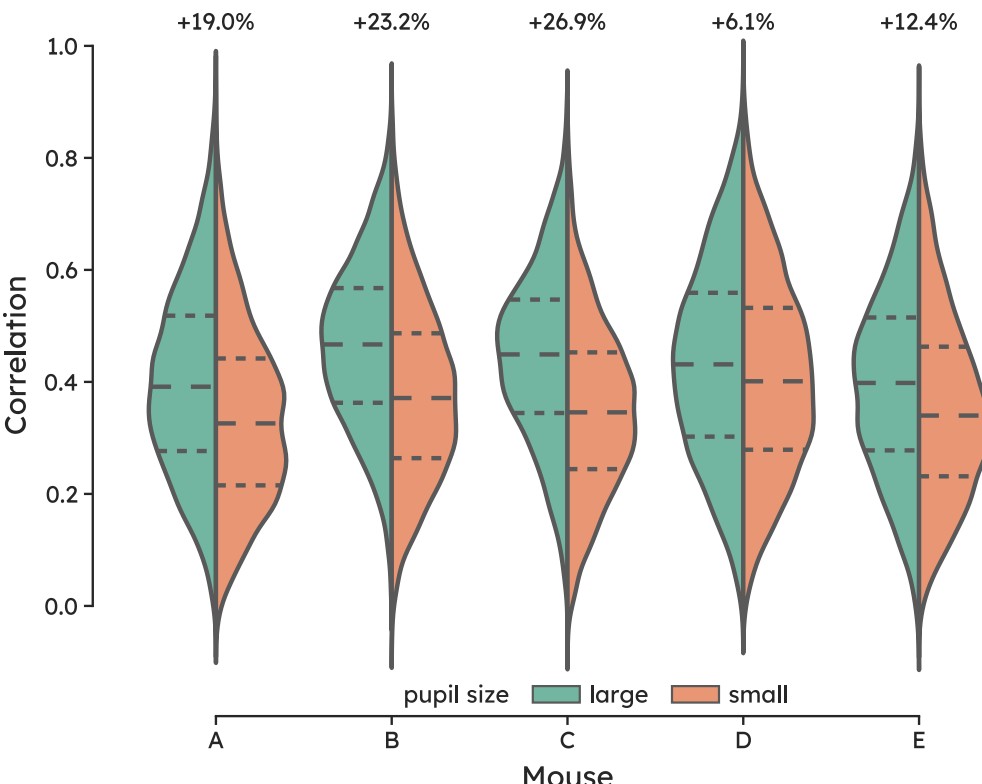

Figure A.4: Predictive performance w.r.t. pupil dilation in DATASET S. Previous work has shown that pupil dilation is an indication of arousal, i.e. stronger (or weaker) neural responses with respect to the visual stimulus (Reimer et al., 2016; Larsen and Waters, 2018). We thus expected a similar tendency could also be observed with our model. Here, we divided the test set into 3 subsets based on pupil dilation. We then compared the predictive performance of the model in the (large) larger third subset against the (small) smaller third subset. We observed that trials with larger pupil sizes are better predicted, with an average difference of +17.5% across animals. The dashed lines indicate the quartiles of the distributions and the percentage above each violin plot shows the relative prediction improvement of the larger set against the smaller set.

## A.8    Readout position and retinotopy

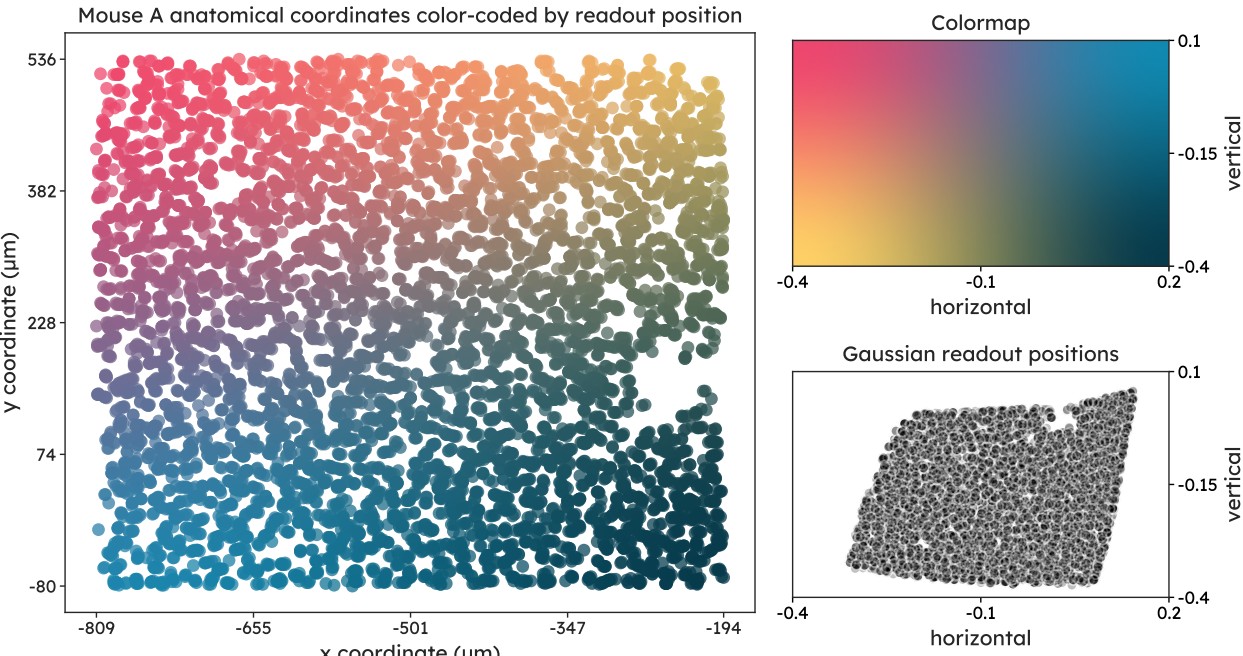

Figure A.5: The learned readout position with respect to neuron anatomical coordinates in MOUSE A. The position network in the Gaussian readout (see Section 3) learns the mapping between the latent visual representation (i.e. output of the core, bottom right panel) and the 2d anatomical location of each neuron (left panel). Lurz et al. (2021) and Willeke et al. (2022) demonstrated that a smooth mapping can be obtained when color-coding each neuron by its corresponding readout position unit. This aligned with our expectation that neurons that are close in space should have a similar receptive field (Garrett et al., 2014). Here, we showed that, despite the substantial architectural change, a similar mapping can also be obtained with the V1T core. The code to generate this plot was written by Willeke et al. (2022) and is available at github.com/sinzlab/sensorium.

