# OpenReview forum: "V1T: large-scale mouse V1 response prediction using a Vision Transformer"
_TMLR — Accepted by TMLR_

### Review · Reviewer_2moi · 2023-06-16

**Summary Of Contributions:**

As the scale of neural data increases, there is an greater pressing need for data-driven modeling approaches that can reliably predict this data and enable “virtual experiments”/ablations to inform future designs. I believe that this work is an important step in this direction, and therefore should be accepted in this venue but with some revisions that I list below (categorized as “Major” and “Minor”).

**Audience:**

Yes

**Broader Impact Concerns:**

No broader impact concern.

**Claims And Evidence:**

Yes

**Requested Changes:**

Major:
To me, the longer-range contribution of the paper is not only the improvement over the baseline CNN, but also that of an unmodified ViT architecture (via the authors’ innovations of finer patch tiling, Gaussian readout, and behavioral MLP) -- as this highlights the need to develop adaptations of (or even completely novel) architectures suited for the particular application of neural-response prediction. Therefore, many of my major suggestions are along this direction:

1.	In Table 1, can you include 3 additional columns: Delta ViT%, p-value CNN, and p-value ViT? (and rename Delta % to Delta CNN%). These would represent the improvement over the baseline ViT (without behavioral variables), along with p-values to indicate significance over both the CNN and ViT.
2.	Would it be possible to try a baseline ViT ensemble as well, and add it as an additional row in the latter half of Table 1?
3.	Figure A.1 is an important figure, and I believe it should be put in the main text, right after Table 1, as it highlights the continued improvement over the baseline CNN as a function of sample size, and greatly enhances the robustness of the main results in Table 1.
4.	Would it be possible to add ViT as an additional bar to the current Figure A.1? It would be very interesting if V1T is more sample efficient than the baseline ViT.


Minor:

1.	On pg. 7, the improvement of using a finer tiling is noted, along with the performance deficit due to LSA. Could these be included as supplementary figures? It would be good to be able refer to them to see the relative performance differences each of these choices leads to. This is the sentence that I am referring to, if it is helpful: “Furthermore, we found that the two efficient tokenizers, SPT and CCT, whose aim is to reduce the number of patches, both failed to improve the model performance, reiterating that a finer tiling of the image is crucial for accurate predictions of cortical activity. Moreover, we found that the LSA attention mechanism, which encourages the model to learn from inter-tokens by masking out the diagonal self-token, led to worse performance, suggesting information from adjacent patches in this task is not as influential as it is in image classification.”
2.	The Introduction on pg. 1 sets up a dichotomy between task-driven and data-driven modeling of neural responses, which given that the improvement over the CNN is not the only interesting aspect of this work, I do not think is necessary (or really warranted). To me, these are just different approaches with different goals. Namely, data-driven models are aiming for maximum predictivity with models tested on the same sort of statistics as what it was trained on – whereas task-driven modelling is aiming for a normative explanation of the evolutionary and developmental constraints of the system (in terms of an architecture, objective function, and training dataset) across a range of conditions, even those that do not share the same statistics as the training dataset. In the revision, it would be good to simply mention that your work is a different goal than understanding ecological constraints of mouse V1, but really to build a predictive model on large-scale datasets and identify core components that can be insightful for designing the next generation of neural data-driven AI models.
3.	Furthermore, related to this last point, there has been a lot of work since Cadena et al. 2019b on much improved, task-driven models of mouse visual cortex (specifically in reference to the sentence on pg. 1: “However, these models do not yield the same [in reference to the primate ventral stream] generalization and prediction results for the mouse visual cortex (Cadena et al. 2019b).”). In particular, the issue with prior task-driven models of mouse visual cortex was that they were “primate-adapted” both in having too deep of an architecture, along with a categorization-centric objective function. However, self-supervised, shallower, and mouse-visual-acuity matched models attain much higher neural predictivity performance, as similarly as deep categorization-optimized CNNs did with the primate ventral stream. Therefore, one relevant work that may be worth citing along these lines is:

A. Nayebi*, N.C.L. Kong*, C. Zhuang, J.L. Gardner, A.M. Norcia, D.L.K. Yamins. “Mouse visual cortex as a limited resource system that self-learns an ecologically-general representation”. bioRxiv 2021. https://www.biorxiv.org/content/10.1101/2021.06.16.448730


**Strengths And Weaknesses:**

See my requested major and minor changes below.

---

> ### Author Response · Authors · 2023-07-07
> **Response to Reviewer 2moi**
>
> We thank the reviewer for their thorough and overall positive review. Below, please find our responses to the questions and concerns raised in the review:
>
> Major changes:
> 1. We thank the reviewer for their suggestion, we added $\Delta$CNN and $\Delta$ViT to Table 1 and Table A.5, and added the p-values between CNN vs V1T and ViT vs V1T in Figure 2 (previously Figure A1).
> 2. We have updated Table 1 with the result of a baseline ViT ensemble model which was fitted in the same fashion as the CNN and V1T ensembles on both datasets. As expected, the ViT ensemble sits between the CNN and V1T ensemble with an average correlation (standard deviation across animals) of 0.424 (0.026) on Dataset S and 0.365 (0.037) on Dataset F.
> 3. We agree with the reviewer that sample efficiency is an important aspect of neural predictive models. We therefore have moved Figure A.1 to the main text in the Results section (now Figure 2).
> 4. We have additionally fitted the ViT with 500, 1500, 2500, 3500, (all) 4500 samples per animal in Dataset S, each with 30 random seeds. With the exception of 500 samples per animal, the ViT core outperforms the CNN model, moreover, ViT continues to improve past 3500 samples per animal. Nevertheless, V1T outperforms ViT with significant difference after 500 samples per animal. The additional results in sample efficiency are shown in Figure 2.
>
> Minor changes:
> 1. We have added Table A.4 to show the best single trial correlation results in Dataset S with all combinations of patch methods (sliding window, 2D convolution, SPT and CCT) and attention mechanisms (original MHA and LSA). Moreover, to highlight the influence of each hyperparameter towards the model prediction performance, we have added Table A.3 to show the hyperparameter importance and correlation with respect to the hyperparameter optimization objective, which were provided by Weights and Biases [1]. Briefly, the importance scores were computed by fitting a random forest with hyperparameter setting and validation performance pairs. The feature importance values from the random forest can then be interpreted as how useful a hyperparameter is in predicting the chosen metric. The linear correlation, on the other hand, shows the correlation between a hyperparameter value and the evaluation metric. More details are available on [Weights & Biases Docs - Parameter Importance](https://docs.wandb.ai/guides/app/features/panels/parameter-importance).
> 2. We thank the reviewer for pointing out that task-driven and data-driven models tackle the neural response modeling with different goals and objectives, we have updated the introduction paragraph as follows:
> “DNN-based models are characterized by two main approaches. On the one hand, task-driven models rely on pre-trained networks optimized on standard vision tasks, such as object recognition, in combination with a readout mechanism to predict neural responses [2, 3, 4]. With the goal of explaining the evolutionary and developmental constraints of the visual system, task-driven models have proven to be successful for predicting visual responses in both primates [4, 5] and mice [6] by obtaining a shared generalized representation of the visual input across animals. On the other hand, data-driven models aim to build a predictive model on large-scale datasets without any assumption on the functional properties of the network. These models share a common representation by being trained end-to-end directly on data from thousands of neurons, and they have been shown to be successful as predictive models for the mouse visual cortex [7, 8]. This approach allows us to identify core components that can be insightful when studying nontrivial computational properties of cortical neurons, especially in combination with experimental verification [9].”
> 3. Please see our response to 2.
>
> [continue]

---

> > ### Author Response · Authors · 2023-07-07
> > **Response to Reviewer 2moi [continue]**
> >
> > We again thank the reviewer for their time and constructive comments to improve this work. We hope that the updated manuscript addresses the concerns raised by the reviewer. We have updated the manuscript accordingly and large changes are written in red font for visibility.
> >
> > [1] Biewald, Lukas. "Experiment tracking with weights and biases." Software available from wandb. com 2 (2020): 233.
> >
> > [2] Yamins, Daniel LK, et al. "Performance-optimized hierarchical models predict neural responses in higher visual cortex." Proceedings of the national academy of sciences 111.23 (2014): 8619-8624.
> >
> > [3] Cadieu, Charles F., et al. "Deep neural networks rival the representation of primate IT cortex for core visual object recognition." PLoS computational biology 10.12 (2014): e1003963.
> >
> > [4] Cadena, Santiago A., et al. "Deep convolutional models improve predictions of macaque V1 responses to natural images." PLoS computational biology 15.4 (2019): e1006897.
> >
> > [5] Yamins, Daniel LK, and James J. DiCarlo. "Using goal-driven deep learning models to understand sensory cortex." Nature neuroscience 19.3 (2016): 356-365.
> >
> > [6] Nayebi, Aran, et al. "Mouse visual cortex as a limited resource system that self-learns an ecologically-general representation." BioRxiv (2022).
> >
> > [7] Lurz, Konstantin-Klemens, et al. "Generalization in data-driven models of primary visual cortex." International Conference on Learning Representations. 2020.
> >
> > [8] Franke, Katrin, et al. "State-dependent pupil dilation rapidly shifts visual feature selectivity." Nature 610.7930 (2022): 128-134.
> >
> > [9] Walker, Edgar Y., et al. "Inception loops discover what excites neurons most using deep predictive models." Nature neuroscience 22.12 (2019): 2060-2065.

---

### Review · Reviewer_TLz3 · 2023-06-22

**Summary Of Contributions:**

This work aims to improve the predictive models of V1 mouse cortex. The authors propose a new architecture for a predictive model that uses a shared ViT backbone and mouse-dependent readouts. The model also incorporates behavioral measures from the pice, such as pupil dilations. This predictive model is trained from scratch using two public datasets. This proposed model is shown to substantially outperform CNN-based models on V1 neuron prediction. Furthermore, the authors show that the incorporation of the behavioral data improves the predictive performance and that the self-attention weights learned by the transformer backbone correlate with the neuron receptive fields.

**Audience:**

Yes

**Claims And Evidence:**

Yes

**Requested Changes:**

I would like the authors to address weaknesses 1 and 2 above.

**Strengths And Weaknesses:**

Overall, I think this is solid work in computational neuroscience. My biggest concern is that it is not clear that a machine learning journal is the most appropriate venue for this work, since the main contributions are in neuroscience. That said, I leave this question to the discretion of the Action Editor and would support publication if the work is judged appropriate for this venue.

Strengths:
The additional analysis showing the relationship between Transformer self-attention weights and neuron receptive field is informative and deeper than the analyses offered by many other works in this area.


Weaknesses:
1. The motivation for why looking at the problem of predicting mouse V1 is important can be strengthened. Even within the field of vision, this is considered quite low-level. The authors can do more to motivate why this is important from a neuroscience point of view, and if they can say something about importance to ML, that would strengthen the reasons to publish this work in an ML venue.
2. On the results side, I would like the authors to report standard deviations of the means every time they report a mean across mice. Otherwise, it's difficult to judge the difference between models.
3. On the modeling side, there are several future avenues for improvement, such as predicting responses for held-out mice, and exploring a pretraining-finetuning paradigm which may be useful to tune the model for a particular dataset. Those are not prerequisites for publication of the current work, though.

---

> ### Author Response · Authors · 2023-07-07
> **Response to Reviewer TLz3**
>
> We thank the reviewer for their overall positive feedback and acknowledging that the analysis on the Transformer self-attention weights and artificial receptive fields are “informative and deeper than the analyses offered by many other works in this area”.
>
> Regarding the suitability for consideration at TMLR, we believe that our work sits in the intersection of computational neuroscience and deep learning, and moreover, we believe this work fulfills the following requirements outlined in the [TMLR submission guidelines](https://www.jmlr.org/tmlr/editorial-policies.html):
> - “computational models of natural learning systems at the behavioral or neural level”
> - “new approaches for analysis, visualization, and understanding of artificial or biological learning systems”
>
> Below, please find our response to the concerns and questions raised in the review:
> 1. ​​Neural response models that can accurately predict neuronal activities in response to natural stimuli allow us to identify core components that can be insightful when studying nontrivial computational properties of cortical neurons, especially in combination with experimental verification [1, 2, 3]. In addition to the relevancy to computational neuroscience, we believe our work contributes two novel ideas: 1) we propose a mechanism to integrate behavioral variables into the core module in order to take advantage of behavioral information and brain state and thus improve visual response predictions, and 2) we investigate how the 'attention weights' learned by the self-attention blocks change with behaviors to gain insight into how the model predicts neural activity. Moreover, from a machine learning perspective, we demonstrate that a ViT-based model, which typically requires large amounts of data to fit, can also be competitive in neural data, even without pre-training. To the best of our knowledge, this is the first ViT-based model to be competitive or even outperform its CNN counterpart in mouse V1 response prediction [4]. We have updated Section 1 Introduction and Section 6 Discussion to highlight the motivation and contribution of this work.
> 2. We have added the standard deviation (in brackets) across animals when the average single trial correlations are reported.
> 3. We thank the reviewer for their suggestion. We have already presented the results on cross-animal and cross-dataset generalization performance in Appendix A.4 (previous Appendix A.2). In the cross-animal setting, (transfer setting) we fitted the models on N-1 animals in Dataset S and fine-tuned the readout module on the N-th animal with the core frozen, (direct setting) we then compared a model (core + readout) that is directly fitted on the N-th animal. We repeated the same process over all 5 animals in Dataset S. Overall, the V1T model outperformed the CNN model in both direct and transfer setting by 3.3% and 6.7%, respectively. Moreover, the V1T model experienced a larger level of performance in the transfer setting as compared to the direct setting (5.6% vs 2.2%).
>   In the cross-dataset setting (transfer), we fitted the models on the gray-scale version of Dataset F, we then fine-tuned the readout modules on animals in Dataset S with the core frozen. Overall, the V1T model outperformed the CNN model in the transfer setting by 4.9%. Nevertheless, the V1T model experienced a larger degradation in prediction performance than the CNN model when compared to the direct training setting (e.g. directly fitting the models on Dataset S) with a drop in performance of -19.0% vs -12.9% in the CNN model.
>
> We thank the reviewer again for their overall positive review and we hope that our responses can address the concerns raised by the reviewer. We have also updated the manuscript to reflect the suggestions made by the reviewer and large changes are written in red font for visibility.
>
> [1] Walker, Edgar Y., et al. "Inception loops discover what excites neurons most using deep predictive models." Nature neuroscience 22.12 (2019): 2060-2065.
>
> [2] Ponce, Carlos R., et al. "Evolving images for visual neurons using a deep generative network reveals coding principles and neuronal preferences." Cell 177.4 (2019): 999-1009.
>
> [3] Willeke, Konstantin F., et al. "Deep learning-driven characterization of single cell tuning in primate visual area V4 unveils topological organization." bioRxiv (2023): 2023-05.
>
> [4] Conwell, Colin, et al. "Neural regression, representational similarity, model zoology & neural taskonomy at scale in rodent visual cortex." Advances in Neural Information Processing Systems 34 (2021): 5590-5607.

---

### Review · Reviewer_fy1h · 2023-06-27

**Summary Of Contributions:**

This work introduces V1T, a novel Vision Transformer-based architecture that predicts neural responses in mouse primary visual cortex to natural images. V1T learns a shared visual and behavioral representation across animals and outperforms previous convolution-based models. V1T also reveals meaningful insights into the visual cortex, such as the correlation between self-attention weights and population receptive fields and the modulation of attention by behavioral variables.

**Audience:**

Yes

**Broader Impact Concerns:**

I do not believe this work requires a broader impact statement.

**Claims And Evidence:**

Yes

**Requested Changes:**

Could the authors elaborate on the distinction between functional and anatomical properties? The authors mention:

"For instance, the center of the self-attention maps learned by our model correlates with the pupil center of the animals, highlighting how features of this architecture do reflect properties of cortical neurons’ receptive fields, in this case, the retinotopy.

and then the authors go on to state:

"This does not imply that our model can directly map onto the anatomy or biophysics of the brain."

Is retinotopy considered a functional or anatomical characteristic of the visual system? This is important because the claims of functional similarity being independent of anatomical or biophysical similarity do not seem possible, especially when you go to lower-level areas of the ventral stream (compared to something further downstream like IT). Can the authors include some clarifying discussion on the statement quoted above?



**Strengths And Weaknesses:**

+ Presents an alternative architecture (V1T) to CNNs that can achieve comparable performance, raising questions about whether CNN architecture is a canonical model or are other factors at play.
+ Shows that behavior as an additional input feature greatly improves prediction performance.
+ Shows some distinct aspects of the receptive field of their proposed V1T model compared to the CNN model.

Questions:

How were the behavioral variables relevant to their area of interest determined? Understanding how these choices interact with prediction performance and, more importantly, what scientific conclusions can be determined from this interaction would be scientifically helpful. Can the authors expand on this interaction?

How are the behavior variables integrated consistently between the different architectures? Were eq (1), (2), (3) the same way behavior information into the CNN architectures? What are ways to control for the fact that it is usually easier to integrate multimodal information in architectures like transformers than CNNs because CNNs features have an explicit spatial dimension to their representations?

"We tried to separate the gradient update for each animal, i.e. one gradient update per core-readout combination, but this led to a significant drop in performance."

Can the authors clarify what the above statement in section 4.2 means? Does this impact the training of the shared core model, or is this specific to the readout layer?

---

> ### Author Response · Authors · 2023-07-07
> **Response to Reviewer fy1h**
>
> We would like to thank the reviewer for their detailed review and overall positive comments. We address the questions and change request raised in the review below.
>
> Questions:
> - The behavioral variables include speed of the animal in the virtual corridor, pupil dilation, dilation derivative and pupil center. For each of these variables, there is prior evidence showing that these variables do affect responses in the primary visual cortex of mice. For instance, Pakan et al. [1] showed that 12% of the recorded V1 neurons decreased their activity with lower running speed, suggesting a clear benefit of considering the speed of the animal for predicting V1 responses. Pupil dilation has been shown to be related to arousal of the animal, with complex modality dependent effects of arousal on the mouse sensory cortex [2]. The pupil center represents the fixation point of the animal and is a proxy for what the animal is paying attention to. As a proof of principle of how a vision transformer can be used to gain insights into the importance of behavioral variables for V1 responses, we showed that the pupil center is correlated to the center of mass of the transformer attention maps.
> We have updated Section 6 Discussion in the manuscript to reflect these points.
> - Franke et al. [3] proposed to integrate behavioral variables as additional channels in the input visual stimulus, i.e. given an input image $x_\text{image} \in \mathbb{R}^{c \times h \times w}$, the $x_\text{behavior} \in \mathbb{R}^{v}$ behavioral information is broadcast to the same spatial dimension as the image: $x_\text{behavior} \in \mathbb{R}^{v \times h \times w}$. The input to the CNN model [4] then concatenates the two vectors and results in an input of shape $\mathbb{R}^{(c+v) \times h \times w}$. The baseline ViT model integrates the behavioral information in the exact same manner (i.e. input the concatenation of image and behaviors $\mathbb{R}^{(c+v) \times h \times w}$). We agree with the reviewer that CNN-based models are more inclined to learn or exploit the spatial relationship in the input which might not be useful in the case of provided behavioral variables. Therefore, one of the main aims of this work, on top of showcasing that the ViT-based model can be a capable V1 neural response predictor, is to introduce an alternative method to incorporate behavioral information. The novel V1T architecture integrates behavioral variables using the formulation detailed in Equation 1 to 3 in a layer-wise fashion, which allows the model to learn different representations of the behavior information in each attention block. We have updated Section 4.1.1 Incorporating behaviors to clarify this point. In addition, Section 3 CNN core and Section 4 Reshape representation are updated to clarify the output shape of various core modules.
> - We observed that updating the core and all readout modules in a single gradient update step via gradient accumulation achieved significantly better performance than updating the core and one readout module at a time, e.g. update core + readout A on mouse A data, then update core + readout B on mouse B data, and so on. We believe this is because the single gradient update step forces the core to learn a shared representation across all animals. However, in the case of a per-animal update scenario, the core representation shifts toward a particular animal at each update step despite the core being shared among all animals.
>
> [continue]

---

> > ### Author Response · Authors · 2023-07-07
> > **Response to Reviewer fy1h [continue]**
> >
> > Request Changes:
> > - We thank the reviewer for highlighting a confusing statement about the relation of functional and anatomical similarity. We did not mean to imply that functional similarity is independent of anatomical or biophysical similarity. Indeed, for the primary visual cortex, retinotopy can be expected to be an anatomical characteristic of the visual system [5]. That said, it is still challenging to map the abstract components of a vision transformer model onto the anatomy of V1, as the architecture of the vision transformer is not designed with biophysical realism in mind.
> >
> >   We changed our statement as follows:
> >
> >   "As a proof of principle of how a vision transformer can be used to gain insights into the importance of behavioral variables for V1 responses, we showed that the center of the self-attention maps learned by our model correlates with the pupil center of the animals, highlighting how features of this architecture do reflect properties of cortical neurons' receptive fields, in this case, the retinotopy. Moreover, our model is able to exploit certain anatomical information, for example the location of neurons within the primary visual cortex, from which we can roughly infer the location of their receptive field since the retinotopic map of mouse primary visual cortex is well characterized [5]. However, while the CNN architecture was inspired by receptive fields of the visual cortex [6], the vision transformer architecture was not and has no direct biological counterpart. Therefore, it is challenging to map the abstract components of a vision transformer onto the anatomy or biophysics of the brain."
> >
> > We again thank the reviewer for their positive review and comments. We have updated the manuscript accordingly and large changes are written in red font for visibility.
> >
> > [1] Pakan, Janelle MP, et al. "The impact of visual cues, reward, and motor feedback on the representation of behaviorally relevant spatial locations in primary visual cortex." Cell reports 24.10 (2018): 2521-2528.
> >
> > [2] Shimaoka, Daisuke, Kenneth D. Harris, and Matteo Carandini. "Effects of arousal on mouse sensory cortex depend on modality." Cell reports 22.12 (2018): 3160-3167.
> >
> > [3] Franke, Katrin, et al. "State-dependent pupil dilation rapidly shifts visual feature selectivity." Nature 610.7930 (2022): 128-134.
> >
> > [4] Lurz, Konstantin-Klemens, et al. "Generalization in data-driven models of primary visual cortex." International Conference on Learning Representations. 2020.
> >
> > [5] Zhuang, J., Ng, L., Williams, D., Valley, M., Li, Y., Garrett, M., & Waters, J. (2017). An extended retinotopic map of mouse cortex. eLife, 6, e18372.
> >
> > [6] Fukushima, K. (1980). A self-organizing neural network model for a mechanism of pattern recognition unaffected by shift in position. Biol, Cybern, 36, 193-202.

---

### Review · Reviewer_9o5e · 2023-06-28

**Summary Of Contributions:**

This paper proposes a new ViT based architecture to predict the mouse V1 neural responses. The architecture integrates visual and behavioral input across animals and surpasses previous CNN models in prediction performance. A detailed analysis of the results from the new architecture and comparison to the previous CNN models are given.


**Audience:**

Yes

**Claims And Evidence:**

Yes

**Requested Changes:**

Although the paper is well written overall, I would like to see the following points discussed in more detail:
1. About hyper parameter tuning:
    1. What is the hyper parameter searching space?
    2. Are the previous CNN models in comparison also tuned for a fair comparison?
2. What is the number of output features in the core module for each of the models? Does this number have any effect on the read-out module?
3. How does the work help understand how the visual system process information?


**Strengths And Weaknesses:**

Strengths:
The paper is clearly written and the analysis is thorough.

Weaknesses:
The biological insight is limited.

---

> ### Author Response · Authors · 2023-07-07
> **Response to Reviewer 9o5e**
>
> We thank the reviewer for their questions and overall positive feedback. Please find our responses and clarification to the questions raised in the review:
>
> 1. Table A.2 details the initial hyperparameter search space for ViT/V1T and final settings. The CNN model configuration was introduced in Lurz et al. [1] and Franke et al. [2] which was selected via an exhaustive Bayesian search [3]. We independently performed another hyperparameter search via Hyperband Bayesian Optimization [4] and could not find a better configuration than the one originally proposed by the authors. We have updated Section 4.2 Training and evaluation to clarify this point.
> 2. We thank the reviewer for pointing out the ambiguity of the output dimension of various core modules. ViT and V1T cores have an output dimension of (C=155, H=29, W=57) and the CNN core has a dimension of (64, 28, 56). As the Gaussian readout module [1] learns to map the visual representation (i.e. core output) to the neuronal response of individual V1 neurons, the dimension of the core output can greatly affect the readout performance. For instance, in the extreme case where the core output has a dimension of (1, 1, 1), the readout module would map all neurons to this single latent space of the visual stimulus, which is not biologically plausible given the size of the recorded cortical area. The dimension of the core output is influenced by the patch size and patch stride in the ViT-based models and filter size, filter stride and padding in the CNN model. In all cases, these configurations were selected via hyperparameter search with the objective of maximizing the single trial correlation in the validation set. We have updated Section 3 CNN core and Section 4.1 Reshape representation with the relevant information.
> 3. Neural response models that can accurately predict neuronal activities in response to natural stimuli allow us to identify core components that can be insightful when studying nontrivial computational properties of cortical neurons, especially in combination with experimental verification [5, 6, 7]. More concretely, we believe our work contributes two novel ideas: 1) we propose a mechanism to integrate behavioral variables into the core module in order to take advantage of behavioral information and brain state and thus improve visual response predictions, and 2) we investigate how the 'attention weights' learned by the self-attention blocks change with behaviors to gain insight into how the model predicts neural activity. The “attention” visualization provides a proof of concept for the validity of the trained model and a framework for further studies investigating the coding of visual information across visual cortical areas. For example, the model could determine which part of the image is processed along the “hierarchy” of visual cortical areas. Higher visual areas are known to have larger receptive fields, so we would expect a larger part of the image to be “relevant” for the prediction. Moreover, our model would be helpful in studies on other cortical areas, for example, those involved in spatial navigation (such as the retrosplenial cortex) that also receive visual inputs, and to determine which part of the image/movie the neurons are reacting to. We have updated Section 1 Introduction and Section 6 Discussion to expand upon these points.
>
> We again would like to thank the reviewer for their time and questions. We hope our responses address the concerns raised in the review. We have updated the manuscript accordingly and major changes are written in red font for visibility.
>
> [1] Lurz, Konstantin-Klemens, et al. "Generalization in data-driven models of primary visual cortex." International Conference on Learning Representations. 2020.
>
> [2] Franke, Katrin, et al. "State-dependent pupil dilation rapidly shifts visual feature selectivity." Nature 610.7930 (2022): 128-134.
>
> [3] Snoek, Jasper, Hugo Larochelle, and Ryan P. Adams. "Practical bayesian optimization of machine learning algorithms." Advances in neural information processing systems 25 (2012).
>
> [4] Li, Lisha, et al. "Hyperband: A novel bandit-based approach to hyperparameter optimization." The Journal of Machine Learning Research 18.1 (2017): 6765-6816.
>
> [5] Walker, Edgar Y., et al. "Inception loops discover what excites neurons most using deep predictive models." Nature neuroscience 22.12 (2019): 2060-2065.
>
> [6] Ponce, Carlos R., et al. "Evolving images for visual neurons using a deep generative network reveals coding principles and neuronal preferences." Cell 177.4 (2019): 999-1009.
>
> [7] Willeke, Konstantin F., et al. "Deep learning-driven characterization of single cell tuning in primate visual area V4 unveils topological organization." bioRxiv (2023): 2023-05.

---

### Public Comment · ~Gehua_Ma1 · 2023-08-25
**Impressive work!**

Very interesting work. And I'd like to know, is there any particular reason behind the choice of the activation function (ELU in this work) in the readout module? There are many options available for activation functions, and theoretically, they can lead to very different behaviors.

---

> ### Author Response · Authors · 2023-08-25
> **Re: Impressive work!**
>
> Thank you for your interest in our work! We have opted to use the same readout module and `ELU + 1` activation combination in order to stay consistent with previous work in this task (namely, Lurz et al. 2021 and Franke et al. 2022) thus allowing us to isolate and compare the effectiveness of the proposed core module. Indeed, the final activation function can lead to different behaviors. We have also experimented with sigmoid activation, where the recorded responses were first normalized to [0, 1], as well as with an exponential activation, though, we found that `ELU + 1` achieved the best performance in the two metrics we used (single trial correlation and average correlation over repeated stimuli).

---

> > ### Public Comment · ~Gehua_Ma1 · 2023-10-02
> > **Thanks for your reply.**
> >
> > Appreciate that.

---

### Decision · Action_Editors · 2023-08-12

**Recommendation:** Accept as is

**Comment:**

This submission proposes a Vision Transformer-based model of mouse primary visual cortex. Reviewers initially asked for further confirmation that the improvements with the proposed changes were not driven by hyperparameter tuning or differences in how behavioral variables were integrated in different architectures, as well as some additional results with baseline ViTs. In their response, the authors provided additional details related to the former two points as well as the requested baselines. Reviewers were satisfied with this response, and all now support acceptance.

**Audience:**

Yes. There is some intersection between TMLR's audience and the neuroscience community, and people at that intersection would be interested in knowing the findings of this paper.

**Claims And Evidence:**

Yes.